# Novel (+)-Neoisopulegol-Based *O*-Benzyl Derivatives as Antimicrobial Agents

**DOI:** 10.3390/ijms22115626

**Published:** 2021-05-26

**Authors:** Tam Minh Le, Thu Huynh, Fatima Zahra Bamou, András Szekeres, Ferenc Fülöp, Zsolt Szakonyi

**Affiliations:** 1Institute of Pharmaceutical Chemistry, University of Szeged, Interdisciplinary Excellent Center, Eötvös utca 6, H-6720 Szeged, Hungary; leminhtam1411@gmail.com (T.M.L.); Bamou.Fatima.Zahra@stud.u-szeged.hu (F.Z.B.); fulop.ferenc@szte.hu (F.F.); 2Stereochemistry Research Group of the Hungarian Academy of Sciences, Eötvös utca 6, H-6720 Szeged, Hungary; 3Department of Microbiology, University of Szeged, Közép fasor 52, 6726 Szeged, Hungary; huynh_thu@hcmut.edu.vn (T.H.); andras.j.szekeres@gmail.com (A.S.); 4Department of Biotecnology, Faculty of Chemical Engineering, Ho Chi Minh University of Technology (HCMUT), 268 Ly Thuong Kiet Street, District 10, Ho Chi Minh City 72607, Vietnam; 5Vietnam National University Ho Chi Minh City, Linh Trung Ward, Thu Duc District, Ho Chi Minh City 71351, Vietnam; 6Interdisciplinary Centre of Natural Products, University of Szeged, Eötvös utca 6, H-6720 Szeged, Hungary

**Keywords:** (+)-neoisopulegol, *O*-Benzyl derivatives, imidazole, 1,2,4-triazole, aminodiol, aminotriol

## Abstract

Discovery of novel antibacterial agents with new structures, which combat pathogens is an urgent task. In this study, a new library of (+)-neoisopulegol-based *O*-benzyl derivatives of aminodiols and aminotriols was designed and synthesized, and their antimicrobial activity against different bacterial and fungal strains were evaluated. The results showed that this new series of synthetic *O*-benzyl compounds exhibit potent antimicrobial activity. Di-*O*-benzyl derivatives showed high activity against Gram-positive bacteria and fungi, but moderate activity against Gram-negative bacteria. Therefore, these compounds may serve a good basis for antibacterial and antifungal drug discovery. Structure–activity relationships were also studied from the aspects of stereochemistry of the *O*-benzyl group on cyclohexane ring and the substituent effects on the ring system.

## 1. Introduction

Heterocyclic compounds, occurring both naturally and produced synthetically, exhibit various pharmacological and biological properties and, therefore, they are interesting synthetic targets in the search of therapeutic agents [1,2]. *O*-Benzyl azole derivatives have played crucial roles in the history of heterocyclic chemistry and have been used extensively as important pharmacophores and synthons in the field of organic chemistry and drug design [1]. Azoles such as imidazole [3] and triazole [4] are the most extensively studied classes of antifungal agents due to their high therapeutic index, good bioavailability, and favorable safety profile [5] while the *O*-benzyl substituent plays an important role in the increased antimicrobial activity of these molecules [6] (Figure 1).

*O*-Benzyl-1,2,4-triazole derivatives were reported to exhibit various pharmacological activities such as antimicrobial [7,8], analgesic [9], anti-inflammatory [10], anticancer [8], antitubercular [11], anti-HIV [12], and antioxidant [13] properties. In addition, drugs with chemotherapeutic effect such as Anastrozole [14] and Letrozole [15] (chemotherapeutic anticancer drug), Ribavirin [16,17,18,19] (antiviral agent), Rizatriptan [20] (antimigraine agent), Alprazolam [21] (anxiolytic agent), Fluconazole [22], and Itraconazole [23] (antifungal agent) as well as Prothioconazole [21] (plant-pathogenic effect) are examples of potent molecules possessing a triazole nucleus [24,25].

*O*-Benzyl imidazole derivatives have evoked considerable attention in recent years because these are endowed with a wide range of pharmaceutical activities. These include antifungal [26], antiparasitic [27], antigiardiasis [28], antitubercular [29], antihistaminic [30], antineuropathic [31], antiobesity [32], antihypertensive [33], antioxidant [34], cardiotonic [35], antithrombotic [36], anti-convulsant [37,38], antiviral [39], and anti-hepatitis B and C virus activity [40] and they may also act as HIV-IPR [41] and IL-1 [42] inhibitors. In particular, a large number of imidazole-based compounds have been widely used drugs such as anticancer [43,44] (dacarbazine, zoledronicacid, azathioprine, and tipifarnib), antifungal [45,46] (clotrimazole, miconazole, ketoconazole, and oxiconazole), antibacterial [47,48] (metronidazole, ornidazole, and secnidazole), antiprotozoal [49,50,51,52,53,54] (megazol, benznidazole, and metronidazole), antihistaminic [55,56,57] (cimetidine, imetit, immepip, and thioperamide), antineuropathic [31,58,59,60,61,62,63,64] (nafimidone, fipamezole, and dexmedetomidine), and antihypertensive [65,66] (losartan, eprosartan, and olmesartan) agents to treat various types of diseases with high therapeutic potency, which shows their huge development value [40].

The increasing number of multidrug-resistant pathogen infections has led to the discovery of new antimicrobial drugs with activity against resistant clinical isolates [67]. In our long-term program toward the synthesis of new antimicrobial agents, we demonstrated that (−)-isopulegol-based *O-*benzyl aminotriol and aminodiol derivatives exert marked antimicrobial effectiveness [68]. Therefore, the present study reports the synthesis of a series of novel (+)-neoisopulegol-based *O*-benzyl derivatives of aminodiols and aminotriols with nitrogen atoms usually incorporated in an imidazole or triazole ring system possessing activity against various bacteria and yeast strains. According to their antimicrobial activities, structure–activity relationships have also been discussed.

## 2. Results

### 2.1. Synthesis of (+)-Neoisopulegol-Based O-Benzyl Derivatives

(+)-Neoisopulegol **2** was prepared from commercially available (−)-isopulegol **1** by oxidizing its hydroxyl function followed by the stereoselective reduction of the resulting carbonyl group applying literature methods [69,70,71,72]. In order to produce *O*-benzyl derivatives, benzyl-protected neoisopulegol **3** was prepared by reacting of **2** with BnBr in the presence of a catalytic amount of KI [73,74]. Without the addition of KI, the reaction proceeded very slowly whereas with the addition of 1 equiv. of KI, the reaction proceeded rapidly due to the formation of more reactive BnI from BnBr [75]. Epoxidation of **3** with *m-*CPBA buffered with Na_2_HPO_4_ provided a 1:2 mixture of epoxides **4a** and **4b** in good yield good yields [76]. The two epoxides were separated by column chromatography to give less polar isomer **4a** and more polar isomer **4b**. Aminolysis of epoxide **4a** with different amines in the presence of LiClO_4_ delivered *O*-benzyl derivatives **5a**–**6a [77,78]**. The role of LiClO_4_ shows enhanced reactivity for the ring opening of epoxides through the coordination of Li^+^ with epoxide oxygen, rendering the epoxide more susceptible to nucleophilic attack by amines, therefore reducing the reaction times dramatically and improved the yields [79,80]. Likewise, no products were observed during ring-opening of the oxirane **3a** with azoles and LiClO_4_. This is probably the difference in reactivity between amines and azoles. Fortunately, it was achieved by reacting **4a** with azoles promoted by K_2_CO_3_ [81]. A possible reaction pathway through potassium carbonate-mediated ring-opening reaction of epoxide **4a** and subsequent nucleophilic addition afforded *O*-benzyl derivatives **7a**–**8a [82]**. Debenzylation of **5a** by hydrogenolysis over Pd/C in MeOH resulted in primary aminodiol **9a** in excellent yield. Since neither aminolysis of the served oxirane **4a** in alkaline condition nor the hydrogenolysis of *N*-benzyl analogue **5a** had an effect on the absolute configuration, the relative configuration of the chiral centers of **5a**–**9a** is known to be the same as that of epoxide **4a** [83,84]. The other epoxide (**4b**) underwent similar reactions to afford **5b**–**9b** in valuable yields (Scheme 1).

To prepare a highly diverse library of *O*-benzyl aminotriols, **3** was oxidized to **10** using SeO_2_/*t*-BuOOH (TBHP) as oxidant [85]. The epoxidation of **10** with *m*-CPBA delivered a 4:1 mixture of epoxides **11a** and **11b**. The separation of **11a** and **11b** was not satisfactory on a gram scale; therefore, the mixture was treated with different nucleophiles resulting in a library of *O*-benzyl derivatives **12**–**15**. In our delight, amine-substituted *O*-benzyl derivatives could easily be separated while in the case of azoles, only the major products were isolated. The debenzylation of **12a** by hydrogenolysis over Pd/C gave primary aminotriol **16a** with good yield (Scheme 2).

During our attempt to improve the separation of epoxides **11a**–**b**, we realized that *O*-benzylation of **10** could serve this purpose. The synthesis of **18a** starting from **10** with NaH/BnBr/KI system, however, provided low-yield transformation (20%). Fortunately, it was achieved starting from **17**, made by the oxidation of **2** [69,70,71,72]. Diol **17** was reacted with benzyl bromide under reflux condition in dry THF to give **18a**, whereas **18b** was prepared at room temperature. Epoxidation of **18a** with *m*-CPBA produced a 1:1 mixture of epoxides **19a** and **19b**. After purification, ring opening of oxiranes **19a**–**b** was accomplished with different nucleophiles resulting in a library of di-*O*-benzyl derivatives **20a**–**24a** and **20b**–**24b**, respectively. The debenzylation of **20a** and **20b** by hydrogenolysis over Pd/C gave, respectively, primary aminotriols **16a** and **16b** in exceptionally high yields (Scheme 3).

The epoxidation of **18b** with *m-*CPBA gave a 3:1 mixture of epoxides **24a** and **24b**. After separation by column chromatography, they were subjected to aminolysis with different nucleophiles to form a library of *O*-benzyl derivatives **25a**–**28a** and **25b**–**28b**, respectively. Primary aminotriols **16a** and **16b** were prepared via the usual way by hydrogenolysis of aminodiols **25a** and **25b** over Pd/C (Scheme 4).

### 2.2. Synthesis of (−)-Isopulegol-Based O-Benzyl Derivatives

Our previous work demonstrated that the *O*-benzyloxy group on the cyclohexyl ring is much more effective to induce antimicrobial activity. Therefore, to explore the role of the configuration of the *O*-benzyloxy group, some (−)-isopulegol-based *O*-benzyl derivatives were also prepared under optimized condition and using literature information [68] (Scheme 5).

### 2.3. Determine Relative Configuration of (+)-Neoisopulegol-Based O-Benzyl Derivatives

Epoxidation of **2** with *t*-BuOOH in the presence of vanadyl acetylacetonate (VO(acac)_2_) as catalyst furnished epoxide **44** in a stereoselective reaction [72]. Debenzylation of **4b** provided **44** in a moderate yield whereas exposure of **44** to NaOH furnished **45** with the retention of stereochemistry [86]. The absolute configuration of *O*-benzyl derivatives **19a** and **25a** was determined by debenzylation together with reduction via hydrogenolysis over Pd/C [87,88] to provide triol **45** with stereochemical retention [68]. The stereochemical structure of epoxide **44** is well-known in the literature [72]; therefore, the absolute configuration of *O*-benzyl derivatives could also be determined (Scheme 6).

### 2.4. Antimicrobial Effects

Since several *O*-benzyl derivatives exerted antimicrobial activities on various microorganisms [68], antimicrobial activities of the prepared *O*-benzyl analogues were also explored against two yeasts as well as two Gram-positive and two Gram-negative bacteria (Table 1, only the best results are shown). Furthermore, the minimal inhibitory concentrations (MIC) of the compounds showed significantly high level (>80%) antimicrobial activity and their MIC values were determined against the test microorganism, where the high inhibition activity was detected (Table 1, in brackets).

## 3. Discussion

### 3.1. Antimicrobial Activity

The MIC values of significant O-benzyl derivatives (I% > 80%) obtained against the tested microorganisms are presented in Table 1. The strongest antifungal activity was shown by compound **22b**, **23a** (di *O*-benzyl aminotriols) at a concentration of 0.78 μg/mL, they were as same as the reference drug ampicillin (0.78 μg/mL). Another di O-benzyl aminotriols **20a** and **39a**–**b** were effective against *B. subtilis* below than 10 μg/mL of MIC values. Moreover, *O*-benzyl aminotriols **5a**–**b**, **7a**–**b**, **31b** together with imidazole-substituted di *O*-benzyl aminotriol **22a** showed lower activity against *B. subtilis* with MIC values in the range between 20 and 50 μg/mL. The weak effect on *B. subtilis* was observed for compounds **3**, **10**, **12a**–**b**, **14a**, **31a**, **42a** (MIC ≥ 100 μg/mL).

Growth inhibition of *S. aureus* was observed at the concentration of 50 μg/mL of *O*-benzyl aminodiols **5a** and **31a**. Imidazole-substituted di *O*-benzyl aminotriol **39b** exhibited relatively high antibacterial potency against *S. aureus* at the MIC values of 3.13 μg/mL, whereas derivatives **7b**, **10**, **12b,** and **14a** was less active against *S. aureus* and inhibited bacterial growth at the concentration of 100 μg/mL. The MICs of standard drug ampicillin for the *S. aureus* were 0.78 μg/mL.

On the other hand, regarding MIC for pathogenic fungi, *O*-benzyl derivatives showed poor activity against all the tested fungal strains, which obtained by the MIC values against *C. albicans* and *C. krusei* (>100 μg/mL).

As shown in Table 1, *N*-benzyl and imidazole-substituted *O*-benzyl derivatives showed significant inhibitory activity against Gram-positive bacteria *B. subtilis* and *S. aureus*. Di-*O*-benzyl-substituted derivatives (**20**, **22**–**23**, **39**–**40**) exerted bactericidal activities against the bacterial species of *B. subtilis* and *S. aureus* at low concentrations (10 μM). Only **12a**–**b** showed significant effect against Gram-negative bacterium *P. aeruginosa* as well as a moderate effect against *E. coli* (**12b**). Other derivatives possessed moderate antibacterial activity against *P. aeruginosa*. Three di-*O*-benzyl derivatives (**20b**, **22b**, **39b**) were highly effective against both *C. albicans* and *C. krusei*. Furthermore, *O*-benzyl derivatives **27b** and **43b** were found to exhibit marked growth inhibition against *C. albicans*. *N*-Dibenzyl-substituted *O*-benzyl derivatives were found to be weakly active or inactive against all tested strains.

The obtained results showed that all synthetic derivatives proved to be more active against Gram-positive than against Gram-negative bacteria. *O*-benzyl derivatives that contain *N-*benzyl and imidazole substitution were the most active compounds against Gram-positive bacteria and had moderate antimicrobial effect against the *P. aeruginosa* (Gram-negative) strain. The mechanism of bactericidal action of heterocycles containing the imidazole ring is thought to be due to disruption of intermolecular interactions in the cell membrane. This can cause dissociation of cellular membrane lipid bilayers, which compromises cellular permeability controls and induces leakage of cellular contents [89]. 

Regarding the yeasts, *N*-benzyl- and imidazole-substituted *O*-benzyl derivatives were also found to be the most active compounds against *C. albicans*. The imidazole derivatives can inhibit the transformation of blastospores of *C. albicans* into the invasive mycelial form [90]. In addition, the preliminary in vitro antifungal screening indicated that *S*-isomers showed better potency compared to *R*-isomers against *C. albicans.* Since the widely accepted primary effect of imidazoles is the inhibition of cytochrome P450-mediated 14a-sterol demethylase of the ergosterol precursor lanosterol from *C. albians* [91]. This enzyme with strict substrate requirements interacted differentially with the stereoisomers of *O*-benzyl derivatives, therefore the affinity of *O*-benzyl derivatives for cytochrome P-450 enzymes involved in steroid synthesis is highly dependent on the stereochemistry of the entire molecule.

The results obtained showed that the tested *O*-benzyl derivatives that contain *N*-dibenzyl substituents have no antibacterial or antifungal activity against any of the tested pathogenic species of bacteria and fungi. The steric hindrance of the substituents, which prevents the destruction of normal permeability, might be the reason for the low antimicrobial and antifungal activity of the *N*-dibenzyl-substituted derivatives. Therefore, the inactivity of *N*-dibenzyl derivatives observed in the present study can be due to the mode of substitution.

### 3.2. Structure-Activity Relationship

(i) *N*,*O*-dibenzyl aminodiols (**5a**–**b**) exhibited significant inhibitory activity against both Gram-positive bacteria (*B. subtilis* and *S. aureus*) and Gram-positive bacteria (*P. aeruginosa* ) as well as yeast (*C. albicans* and *C. krusei*). Replacing *N*-benzyl substitution by imidazole (**7a**–**b**) led to the loss of activity against *C. krusei.*

(ii) When the -CH_3_ group of isopropyl part was changed to -CH_2_OH, disappearance on inhibitory activity against *S. aureus* and *C. krusei* was observed on *N*,*O*-dibenzyl aminodiol containing *R*-isomer (**12a**) whereas the other stereoisomer (**12b**) exhibited an additive effect on *E. coli*. In the case of imidazole *O*-benzyl aminotriols, this route reduced activity on *C. albicans* with *R*-isomer (**14a**) and totally lost on antifungal effectiveness on the other isomer (**14b**).

(iii) Benzylation of -CH_2_OH provided di *O*-benzyl aminotriols. Our tests revealed that the lack of antifungal activity and high potency against positive-Gram bacteria in both *N*-benzyl (**20a**–**b**) and imidazole (**24a**–**b**) aminotriols were produced at a low concentration (10 µM). This modification probably improves the lipophilic properties that enhanced interactions in the cell membrane. In addition, the synthesized triazole analogues (**23a**–**b**) also exhibit marked growth inhibition against Gram-positive bacteria (*B. subtilis* and *S. aureus*) and Gram-positive bacteria (*P. aeruginosa*).

(iv) The almost complete loss of antimicrobial activity resulting from the debenzylation on the cyclohexane ring demonstrated with aminotriol derivatives (**25a**–**b**) suggests that the benzyl moiety on cyclohexyl ring is a key element to have satisfactory antimicrobial activity in the case of *N*,*O*-dibenzyl aminotriol whereas they exert markedly selective antibacterial action on *P. aeruginosa* in the case of imidazole *O*-benzyl aminotriol.

(v) In the stereochemistry study of the OH group on the cyclohexyl ring, aminodiol with *S*-configuration (**27a**–**b**) displayed a potential negative-Gram bacterial effect (*P. aeruginosa)* while derivatives with *R*-configuration (**42a**–**b**) had significant positive-Gram bacterial effect (*B. subtilis*) whereas the stereochemistry of the *O*-benzyl substituent on the cyclohexane ring in the aminodiol and aminotriol function has no influence on the antimicrobial effect. 

(vi) The available data demonstrated that most of the *N*-benzyl and imidazole-substituted *O*-benzyl derivatives exhibited more antimicrobial potency than triazole or *N*,*N*-dibenzyl *O*-benzyl ones.

(vii) Further, this result indicates that *S*-isomer showed better potency compared to *R*-isomer against fungi.

## 4. Materials and Methods

### 4.1. General Methods

Commercially available compounds were used as obtained from suppliers (Molar Chemicals Ltd., Halásztelek, Hungary; Merck Ltd., Budapest, Hungary and VWR International Ltd., Debrecen, Hungary), while solvents were dried according to standard procedures. Optical rotations were measured in MeOH at 20 °C, with a Perkin-Elmer 341 polarimeter (PerkinElmer Inc., Shelton, CT, USA). Chromatographic separations and monitoring of reactions were carried out on Merck Kieselgel 60 (Merck Ltd., Budapest, Hungary). Elemental analyses for all prepared compounds were performed on a Perkin-Elmer 2400 Elemental Analyzer (PerkinElmer Inc., Waltham, MA, USA). GC measurements for direct separation of commercially available enantiomers of isopulegol to determine the enantiomeric purity of starting material **1** were performed on a Chirasil-DEX CB column (2500 × 0.25 mm I.D.) on a Perkin-Elmer Autosystem XL GC equipped with a Flame Ionization Detector (Perkin-Elmer Corporation, Norwalk, CT, USA) and a Turbochrom Workstation data system (Perkin-Elmer Corp., Norwalk, CT, USA). Melting points were determined on a Kofler apparatus (Nagema, Dresden, Germany) and are uncorrected. ^1^H- and ^13^C-NMR spectra were recorded on Brucker Avance DRX 500 spectrometer (Bruker Biospin, Karlsruhe, Baden Württemberg, Germany) [500 MHz (^1^H) and 125 MHz (^13^C), δ = 0 (TMS)]. Chemical shifts are expressed in ppm (δ) relative to TMS as the internal reference. *J* values are given by Hz.

(−)-Isopulegol (**1**) is commercially available from Merck Co with *ee* = 95%, (${\left[\alpha \right]}_{D}^{20}$ = −22.0, neat) and its enatimomer (+)-**1** (*ee* = 90%, [${\left[\alpha \right]}_{D}^{20}$ = +22.0, neat). (+)-Neoisopulegol (**2**) (${\left[\alpha \right]}_{D}^{20}$ = +28.7, c = 17.2, CHCl_3_) and its enatimomer (−)-**2** (${\left[\alpha \right]}_{D}^{20}$ = −22.2, c = 2.0, CHCl_3_) were synthesized from (−)-**1** and its isomer (+)-**1** following a reported procedure, respectively [71]. Diol **17**, epoxide **44** [72] as well as compounds **29**, **33,** and **37a**–**b** [68] were prepared according to literature procedures. All spectroscopic data were similar to those described therein. Since any of the applied transformations do not reach all the four chiral centers at the same time, giving rise to racemization, rather only the formation of the prescribed and isolated diastereoisomers, we believe that the enantiomer purity of the prepared compounds can be defined as *ee* ≥ 95% (commercial (−)-isopulegol). ^1^H, ^13^C, HSQC, HMBC and NOESY NMR spectra of new compounds and GC chromatograms of isopulegol enantiomers are available in Appendix A.

### 4.2. Experimental Section and Compound Characterisations

#### 4.2.1. (*S*)-2-((1*R*,2*R*,4*R*)-2-Hydroxy-4-methylcyclohexyl)propane-1,2-diol (**45**)

Compound **44** (0.60 mmol) was treated with DMSO (3.0 mL) and 3 M NaOH (3.0 mL). The resulting homogenous solution was stirred at 80 °C for 2 h. After being cooled to room temperature, EtOAc (20 mL) was added, and the aqueous layer was washed with EtOAc (3 × 20 mL). The combined organic layers were dried over Na_2_SO_4_, filtered, and concentrated in vacuo. The crude material was purified by column chromatography on silica gel (*n*-hexane:EtOAc = 1:4) to provide compound **45**.

Yield: 76%, colorless oil. ${\left[\alpha \right]}_{D}^{20}$ = +14.0 (c 0.22, MeOH). ^1^H NMR (500 MHz, CDCl_3_): δ = 0.88 (3H, d, *J* = 6.2 Hz), 0.91–0.97 (1H, m), 1.10–1.16 (1H, m), 1.25 (3H, s), 1.35–1.39 (1H, m), 1.49–1.53 (1H, m), 1.62–1.70 (1H, m), 1.76–1.85 (3H, m), 3.23 (2H, brs), 3.29 (1H, d, *J* = 11.1 Hz), 3.63 (1H, d, *J* = 11.1 Hz), 4.38 (1H, s). ^13^C NMR (125 MHz, CDCl_3_): δ = 21.4, 22.3, 25.0, 25.9, 35.2, 42.8, 48.9, 67.0, 67.3, 74.4. Found: C, 63.83; H, 10.69. Anal. Calcd for C_10_H_20_O_3_: C, 63.80; H, 10.71.

#### 4.2.2. 2-((1*S*,2*S*,4*R*)-2-(Benzyloxy)-4-methylcyclohexyl)prop-2-en-1-ol (**10**)

To a solution of *t*-BuOOH (70% purity in H_2_O, 32.80 mmol) in CH_2_Cl_2_ (50 mL), dried briefly (Na_2_SO_4_), was added finely powdered SeO_2_ (1.96 mmol) followed by 30 minutes by the addition of **3** (8.20 mmol). After stirring for 20 h at 25 °C, saturated NaHCO_3_ solution (50 mL) was added, then CH_2_Cl_2_ phases were separated and the aqueous phase was extracted with CH_2_Cl_2_ (3 × 50 mL). The organic layer was dried (Na_2_SO_4_) and concentrated in vacuo to afford colorless oil, which was added at 0 °C to a suspension of NaBH_4_ (24.60 mmol) in dry MeOH (50 mL). The reaction mixture was stirred for 2 h at 0 °C while the reaction progress was monitored by TLC. When the reaction was complete, the mixture was poured into brine (100 mL) and the product was extracted with CH_2_Cl_2_ (3 × 100 mL). The combined extracts were washed with water and dried over anhydrous Na_2_SO_4_. The solvent was evaporated in vacuo. The crude product was purified by column chromatography on silica gel using *n*-hexane:EtOAc = 4:1.

Yield: 27%, colorless oil. ${\left[\alpha \right]}_{D}^{20}$ = +29.0 (c 0.27, MeOH). ^1^H NMR (500 MHz, CDCl_3_): δ = 0.89 (3H, d, *J* = 6.4 Hz), 0.94–1.07 (2H, m), 1.50–1.55 (1H, m), 1.76–1.80 (2H, m), 1.87–1.95 (1H, m), 2.07–2.11 (1H, m), 2.24 (1H, d, *J* = 13.0 Hz), 2.67 (1H, t, *J* = 5.4 Hz), 3.71 (1H, d, *J* = 2.4 Hz), 3.94 (1H, dd, *J* = 12.7, 5.8 Hz), 4.06 (1H, dd, *J* = 12.7, 4.1 Hz), 4.34 (1H, d, *J* = 11.6 Hz), 4.60 (1H, d, *J* = 11.7 Hz), 4.96 (1H, s), 5.07 (1H, d, *J* = 1.0 Hz), 7.25–7.32 (5H, m). ^13^C NMR (125 MHz, CDCl_3_): δ = 22.5, 25.0, 26.0, 35.0, 37.5, 46.6, 65.2, 70.6, 77.3, 113.2, 127.7, 127.9, 128.4, 138.4, 151.0. Found: C, 78.40; H, 9.33. Anal. Calcd for C_17_H_24_O_2_: C, 78.42; H, 9.29.

#### 4.2.3. General Procedure for Benzylation

A suspension of NaH (60% purity, 6.6 mmol) in dry THF (10 mL) was added to a solution of alcohol (6.6 mmol) in dry THF (20 mL). The reaction mixture was stirred at 25 °C for 30 min before benzyl bromide (9.9–19.8 mmol) and KI (6.6 mmol) were added to the mixture. Stirring was continued for 12–24 h at 25–60 °C. When the reaction was complete, the mixture was poured into saturated NH_4_Cl solution (30 mL) and extracted with EtOAc (3 × 50 mL). The combined organic phase was dried over anhydrous Na_2_SO_4_. The solvent was evaporated in vacuo and the crude product was purified by column chromatography on silica gel to provide **3** or **18a**–**b**, respectively.

##### ((((1*S*,2*S*,5*R*)-5-Methyl-2-(prop-1-en-2-yl)cyclohexyl)oxy)methyl)benzene (**3**)

Prepared with **2** and benzyl bromide (9.9 mmol) at reflux for 12 h and eluted by *n*-hexane:EtOAc = 19:1. Yield: 63%, colorless oil. ${\left[\alpha \right]}_{D}^{20}$ = +24.0 (c 0.28, MeOH). ^1^H NMR (500 MHz, CDCl_3_): δ = 0.87 (3H, d, *J* = 6.4 Hz), 0.86–0.89 (2H, m), 0.92–1.00 (2H, m), 1.25–1.31 (2H, m), 1.51–1.54 (1H, m), 1.73 (3H, s), 1.74–1.80 (2H, m), 1.85–1.95 (2H, m), 2.01–2.06 (1H, m), 3.75 (1H, d, *J* = 1.6 Hz), 4.38 (1H, d, *J* = 12.1 Hz), 4.56 (1H, d, *J* = 12.1 Hz), 4.77 (1H, d, *J* = 0.5 Hz), 4.80 (1H, s), 7.21–7.32 (5H, m). ^13^C NMR (125 MHz, CDCl_3_): δ = 22.4, 22.5, 22.8, 25.2, 26.3, 35.2, 38.6, 48.6, 70.8, 76.1, 110.5, 127.2, 127.5, 128.2, 139.8, 148.0. Found: C, 83.50; H, 9.93. Anal. Calcd for C_17_H_24_O: C, 83.55; H, 9.90.

##### (((2-((1*S*,2*S*,4*R*)-2-(Benzyloxy)-4-methylcyclohexyl)allyl)oxy)methyl)benzene (**18a**)

Prepared with **17** and benzyl bromide (19.8 mmol) at reflux for 24 h and eluted by *n*-hexane:EtOAc = 19:1. Yield: 56%, colorless oil. ${\left[\alpha \right]}_{D}^{20}$ = +20.0 (c 0.25, MeOH). ^1^H NMR (500 MHz, CDCl_3_): δ = 0.88 (3H, d, *J* = 6.4 Hz), 0.94–1.01 (2H, m), 1.20–1.30 (3H, m), 1.52–1.57 (5H, m), 1.75–1.78 (2H, m), 1.83–1.91 (1H, m), 2.02–2.05 (1H, m), 2.13–2.17 (1H, m), 3.71 (1H, s), 3.89 (1H, d, *J* = 12.5 Hz), 3.99 (1H, d, *J* = 12.5 Hz), 4.31 (1H, d, *J* = 12.0 Hz), 4.38 (1H, d, *J* = 11.9 Hz), 4.46 (1H, d, *J* = 12.0 Hz), 4.54 (1H, d, *J* = 12.0 Hz), 5.06 (1H, s), 5.14 (1H, s), 7.23–7.36 (10H, m). ^13^C NMR (125 MHz, CDCl_3_): δ = 22.5, 25.2, 26.3, 35.2, 38.2, 44.5, 70.6, 72.0, 73.2, 112.9, 127.3, 127.5, 127.6, 127.8, 128.3, 128.5, 147.8. Found: C, 82.27; H, 8.67. Anal. Calcd for C_24_H_30_O_2_: C, 82.24; H, 8.63.

##### (1*S*,2*S*,5*R*)-2-(3-(Benzyloxy)prop-1-en-2-yl)-5-methylcyclohexanol (**18b**)

Prepared with **17** and benzyl bromide (9.9 mmol) at 25 °C for 12 h and eluted by *n*-hexane:EtOAc = 9:1. Yield: 59%, colorless oil. ${\left[\alpha \right]}_{D}^{20}$ = +33.0 (c 0.28, MeOH). ^1^H NMR (500 MHz, CDCl_3_): δ = 0.88 (3H, d, *J* = 6.5 Hz), 0.91–1.01 (1H, m), 1.13 (1H, t, *J* = 12.9 Hz), 1.41–1.47 (1H, m), 1.62 (1H, s), 1.74–1.83 (3H, m), 1.90–1.95 (1H, m), 2.21 (1H, d, *J* = 12.7 Hz), 2.26 (1H, s), 3.91 (1H, d, *J* = 11.8 Hz), 3.96 (1H, s), 4.07 (1H, d, *J* = 11.7 Hz), 4.48 (1H, d, *J* = 11.9 Hz), 4.54 (1H, d, *J* = 11.8 Hz), 5.06 (1H, s), 5.21 (1H, s), 7.25–7.36 (5H, m). ^13^C NMR (125 MHz, CDCl_3_): δ = 22.4, 24.1, 25.8, 35.0, 41.3, 45.9, 67.7, 72.5, 72.7, 115.2, 127.9, 128.6, 138.0, 143.4, 147.8. Found: C, 78.45; H, 9.27. Anal. Calcd for C_17_H_24_O_2_: C, 78.42; H, 9.29.

#### 4.2.4. General Procedure of Epoxidation

To the solution of allylic alcohol derivatives (11.9 mmol) in CH_2_Cl_2_ (50 mL), Na_2_HPO_4_·12H_2_O (35.7 mmol) in water (130 mL) and *m*-CPBA (70% purity, 23.8 mmol) were added at 0 °C, then the mixture was stirred at room temperature. When the reaction was complete (2 h), the mixture was separated, and the aqueous phase was extracted with CH_2_Cl_2_ (100 mL). The organic layer was washed with 5% KOH solution (3 × 50 mL), dried (Na_2_SO_4_) and concentrated in vacuo. The residue was purified by column chromatography on silica gel with an appropriate solvent mixture to afford epoxides.

##### (*R*)-2-((1*R*,2*S*,4*R*)-2-(benzyloxy)-4-methylcyclohexyl)-2-methyloxirane (**4a**)

Prepared with **3** eluted by *n*-hexane:EtOAc = 9:1. Yield: 23%, colorless oil. ${\left[\alpha \right]}_{D}^{20}$ = +32.0 (c 0.285, MeOH). ^1^H NMR (500 MHz, CDCl_3_): δ = 0.87 (3H, d, *J* = 6.5 Hz), 0.85–0.95 (2H, m), 1.28 (3H, s), 1.44–1.56 (3H, m), 1.71–1.76 (2H, m), 2.06–2.11 (1H, m), 2.51 (1H, d, *J* = 4.9 Hz), 2.73 (1H, d, *J* = 4.9 Hz), 3.87 (1H, d, *J* = 2.1 Hz), 4.39 (1H, d, *J* = 11.8 Hz), 4.62 (1H, d, *J* = 11.8 Hz), 7.25–7.33 (5H, m). ^13^C NMR (125 MHz, CDCl_3_): δ = 20.8, 22.1, 22.4, 26.4, 34.6, 37.7, 47.2, 53.6, 58.5, 70.3, 74.9, 127.4, 127.5, 128.4, 139.4. Found: C, 78.47; H 9.33. Anal. Calcd for C_17_H_24_O_2_: C, 78.42; H, 9.29.

##### (*S*)-2-((1*R*,2*S*,4*R*)-2-(Benzyloxy)-4-methylcyclohexyl)-2-methyloxirane (**4b**)

Prepared with **3** eluted by *n*-hexane:EtOAc = 9:1. Yield: 47%, colorless oil. ${\left[\alpha \right]}_{D}^{20}$ = +88.7 (c 0.385, MeOH). ^1^H NMR (500 MHz, CDCl_3_): δ = 0.87 (3H, d, *J* = 6.4 Hz), 0.84–0.95 (2H, m), 1.19–1.24 (1H, m), 1.33 (3H, s), 1.62–1.66 (1H, m), 1.77–1.82 (3H, m), 2.02–2.07 (1H, m), 2.49 (1H, d, *J* = 4.9 Hz), 2.68 (1H, d, *J* = 4.9 Hz), 2.72 (1H, d, *J* = 2.2 Hz), 4.35 (1H, d, *J* = 11.7 Hz), 4.60 (1H, d, *J* = 11.7 Hz), 7.25–7.34 (5H, m). ^13^C NMR (125 MHz, CDCl_3_): δ = 20.5, 22.4, 23.2, 26.3, 34.8, 37.6, 53.4, 59.4, 70.1, 75.6, 76.9, 77.1, 127.4, 127.6, 128.4, 139.2. Found: C, 78.40; H 9.25. Anal. Calcd for C_17_H_24_O_2_: C, 78.42; H, 9.29.

##### (*S*)-2-((1*R*,2*S*,4*R*)-2-(Benzyloxy)-4-methylcyclohexyl)-2-((benzyloxy)methyl)oxirane (**19a**)

Prepared with **18a** eluted by *n*-hexane:EtOAc = 9:1. Yield: 36%, colorless oil. ${\left[\alpha \right]}_{D}^{20}$ = +47.0 (c 0.25, MeOH). ^1^H NMR (500 MHz, CDCl_3_): δ = 0.87 (3H, d, *J* = 6.4 Hz), 0.86–0.97 (2H, m), 1.40–1.50 (2H, m), 1.69–1.75 (2H, m), 1.89–1.93 (1H, m), 2.04–2.09 (1H, m), 2.64 (1H, d, *J* = 4.7 Hz), 2.80 (1H, d, *J* = 4.8 Hz), 2.87 (1H, d, *J* = 11.6 Hz), 3.73 (1H, d, *J* = 11.6 Hz), 3.83 (1H, d, *J* = 5.4 Hz), 4.25 (1H, d, *J* = 11.8 Hz), 4.40 (1H, d, *J* = 12.1 Hz), 4.51 (1H, d, *J* = 12.0 Hz), 4.57 (1H, d, *J* = 11.9 Hz), 7.24–7.34 (10H, m). ^13^C NMR (125 MHz, CDCl_3_): δ = 21.1, 22.4, 26.4, 34.5, 37.3, 42.4, 48.3, 60.1, 70.1, 71.6, 73.2, 74.3, 127.4, 127.5, 127.7, 127.8, 128.4, 128.5. Found: C, 78.67; H 8.23. Anal. Calcd for C_24_H_30_O_3_: C, 78.65; H, 8.25.

##### (*R*)-2-((1*R*,2*S*,4*R*)-2-(Benzyloxy)-4-methylcyclohexyl)-2-((benzyloxy)methyl)oxirane (**19b**)

Prepared with **18a** eluted by *n*-hexane:EtOAc = 9:1. Yield: 36%, colorless oil. ${\left[\alpha \right]}_{D}^{20}$ = +54.0 (c 0.25, MeOH). ^1^H NMR (500 MHz, CDCl_3_): δ = 0.86 (3H, d, *J* = 6.5 Hz), 0.85–0.95 (4H, m), 1.26–1.29 (2H, m), 1.50 (2H, m), 1.62–1.76 (5H, m), 2.01–2.04 (1H, m), 2.69 (1H, d, *J* = 5.3 Hz), 2.84 (1H, d, *J* = 5.4 Hz), 3.58 (1H, d, *J* = 10.9 Hz), 3.70 (1H, d, *J* = 11.4 Hz), 3.76 (1H, s) 4.32 (1H, d, *J* = 11.6 Hz), 4.48 (2H, s), 4.54 (1H, d, *J* = 11.6 Hz), 7.23–7.32 (10H, m). ^13^C NMR (125 MHz, CDCl_3_): δ = 22.4, 23.1, 26.3, 34.9, 37.7, 43.9, 48.9, 60.8, 70.2, 71.3, 73.5, 74.8, 127.4, 127.7, 127.8, 128.4, 128.4, 128.5, 138.6, 139.3. Found: C, 78.62; H 8.23. Anal. Calcd for C_24_H_30_O_3_: C, 78.65; H, 8.25.

##### (1*R*,2*R*,5*R*)-2-((*S*)-2-((Benzyloxy)methyl)oxiran-2-yl)-5-methylcyclohexanol (**24a**)

Prepared with **18b** eluted by *n*-hexane:EtOAc = 4:1. Yield: 42%, colorless oil. ${\left[\alpha \right]}_{D}^{20}$ = +37.0 (c 0.275, MeOH). ^1^H NMR (500 MHz, CDCl_3_): δ = 0.86 (3H, d, *J =* 6.4 Hz), 0.88–0.96 (1H, m), 1.00–1.06 (1H, m), 1.45–1.49 (1H, m), 1.55–1.59 (2H, m), 1.66–1.1.78 (2H, m), 1.82–1.87 (2H, m), 2.67 (1H, d, *J* = 4.6 Hz), 2.80 (1H, d, *J* = 4.6 Hz), 3.22 (1H, d, *J* = 10.3 Hz), 3.37 (1H, s), 3.82 (1H, d, *J* = 10.3 Hz), 4.18 (1H, s), 4.53 (1H, d, *J* = 11.6 Hz), 4.57 (1H, d, *J* = 11.8 Hz), 7.25–7.37 (5H, m). ^13^C NMR (125 MHz, CDCl_3_): δ = 22.2, 22.3, 25.6, 34.7, 41.7, 44.0, 50.2, 60.6, 67.8, 72.1, 73.7, 127.9, 128.0, 128.5, 137.1. Found: C, 78.90; H 8.77. Anal. Calcd for C_17_H_24_O_3_: C, 73.88; H, 8.75.

##### (1*R*,2*R*,5*R*)-2-((*R*)-2-((Benzyloxy)methyl)oxiran-2-yl)-5-methylcyclohexanol (**24b**)

Prepared with **18b** eluted by *n*-hexane:EtOAc = 4:1. Yield: 15%, colorless oil. ${\left[\alpha \right]}_{D}^{20}$ = +24.0 (c 0.295, MeOH). ^1^H NMR (500 MHz, CDCl_3_): δ = 0.86 (3H, d, *J* = 6.4 Hz), 0.88–0.95 (1H, m), 1.02–1.07 (1H, m), 1.47–1.50 (1H, m), 1.57 (1H, s), 1.59–1.66 (2H, m), 1.74–1.77 (1H, m), 1.82–1.88 (2H, m), 2.69 (1H, d, *J* = 4.6 Hz), 2.85 (1H, d, *J* = 4.6 Hz), 3.24 (1H, s), 3.43 (1H, d, *J* = 10.8 Hz), 3.69 (1H, d, *J* = 10.9 Hz), 4.14 (1H, s), 4.53 (1H, d, *J* = 11.8 Hz), 4.61 (1H, d, *J* = 11.9 Hz), 7.25–7.35 (5H, m). ^13^C NMR (125 MHz, CDCl_3_): δ = 21.9, 22.3, 25.9, 34.9, 41.8, 44.3, 50.7, 66.7, 72.2, 73.8, 128.0, 128.1, 128.7, 137.4. Found: C, 78.85; H 8.74. Anal. Calcd for C_17_H_24_O_3_: C, 73.88; H, 8.75.

#### 4.2.5. General Procedure for Ring-Opening of Epoxides with Different Amines

A solution of epoxides (2.9 mmol) in MeCN (30 mL) was added to the appropriate amines (5.8 mmol) in MeCN (10 mL) and LiClO_4_ (2.9 mmol). The mixture was kept at reflux temperature for 6–20 h. When the reaction was completed (indicated by TLC), the mixture was evaporated to dryness, the residue was again dissolved in water (15 mL), and then extracted with CH_2_Cl_2_ (3 × 50 mL). The combined organic phase was dried (Na_2_SO_4_), filtered, and concentrated. The crude product was purified by column chromatography on silica gel with an appropriate solvent mixture, resulting in *O*-benzyl derivatives, respectively.

##### (*R*)-1-(Benzylamino)-2-((1*R*,2*S*,4*R*)-2-(benzyloxy)-4-methylcyclohexyl)propan-2-ol (**5a**)

Prepared with **4a** with benzylamine at reflux for 20 h and eluted by *n*-hexane:EtOAc = 1:1. Yield: 78%, colorless oil. ${\left[\alpha \right]}_{D}^{20}$ = +41.0 (c 0.275, MeOH). ^1^H NMR (500 MHz, CDCl_3_): δ = 0.87 (3H, d, *J* = 6.6 Hz), 0.83–0.97 (2H, m), 1.17 (3H, s), 1.42–1.46 (1H, m), 1.62–1.80 (4H, m), 2.05–2.09 (1H, m), 2.54 (1H, d, *J* = 11.6 Hz), 2.63 (1H, d, *J* = 11.6 Hz), 3.71 (1H, d, *J* = 13.3 Hz), 3.80 (1H, d, *J* = 13.3 Hz), 3.92 (1H, d, *J* = 1.5 Hz), 4.13 (1H, d, *J* = 11.3 Hz), 4.50 (1H, d, *J* = 11.2 Hz), 7.23–7.33 (10H, m). ^13^C NMR (125 MHz, CDCl_3_): δ = 21.2, 22.4, 24.3, 26.1, 35.1, 37.1, 46.6, 54.5, 58.2, 69.7, 73.8, 75.5, 127.1, 127.8, 127.9, 128.5, 128.6, 138.4, 140.4. Found: C, 78.45; H, 9.07; N, 3.79. Anal. Calcd for C_24_H_33_NO_2_: C, 78.43; H, 9.05; N, 3.81.

##### (*S*)-1-(Benzylamino)-2-((1*R*,2*S*,4*R*)-2-(benzyloxy)-4-methylcyclohexyl)propan-2-ol (**5b**)

Prepared with **4** with benzylamine at reflux for 20 h and eluted by *n*-hexane:EtOAc = 1:1. Yield: 64%, colorless oil. ${\left[\alpha \right]}_{D}^{20}$ = +7.0 (c 0.25, MeOH). ^1^H NMR (500 MHz, CDCl_3_): δ = 0.90 (3H, d, *J* = 6.4 Hz), 0.85–0.93 (2H, m), 0.93–1.00 (1H, m), 1.02–1.07 (1H, m), 1.20–1.29 (4H, m), 1.26 (3H, s), 1.53–1.57 (1H, m), 1.63–1.71 (2H, m), 2.16–2.22 (1H, m), 2.58 (1H, d, *J* = 12.4 Hz), 2.78 (1H, d, *J* = 12.4 Hz), 3.53 (1H, d, *J* = 13.4 Hz), 3.63 (1H, d, *J* = 13.4 Hz), 4.20 (1H, s), 4.37 (1H, d, *J* = 10.3 Hz), 4.63 (1H, d, *J* = 10.4 Hz), 7.02–7.04 (2H, m), 7.28–7.45 (8H, m). ^13^C NMR (125 MHz, CDCl_3_): δ = 22.2, 26.2, 28.1, 29.8, 35.0, 36.9, 49.9, 52.0, 52.7, 70.8, 72.3, 75.0, 128.8, 129.0, 129.1, 129.3, 129.5, 137.2. Found: C, 78.40; H, 9.03; N, 3.84. Anal. Calcd for C_24_H_33_NO_2_: C, 78.43; H, 9.05; N, 3.81.

##### (*R*)-2-((1*R*,2*S*,4*R*)-2-(Benzyloxy)-4-methylcyclohexyl)-1-(dibenzylamino)propan-2-ol (**6a**)

Prepared with **4** with dibenzylamine at reflux for 20 h and eluted by *n*-hexane:EtOAc = 9:1. Yield: 50%, white crystal, m.p = 138–140 °C. ${\left[\alpha \right]}_{D}^{20}$ = +30.0 (c 0.27, MeOH). ^1^H NMR (500 MHz, CDCl_3_): δ = 0.83–0.92 (2H, m), 0.84 (3H, d, *J* = 6.5 Hz), 1.13 (3H, s), 1.53–1.56 (4H, m), 1.63–1.75 (3H, m), 2.01 (1H, dd, *J* = 14.1, 1.9 Hz), 2.86 (1H, d, *J* = 13.8 Hz), 3.3. (1H, dd, *J* = 13.6, 4.9 Hz), 3.67 (2H, d, *J* = 12.4 Hz), 4.06 (1H, d, *J* = 11.9 Hz), 4.43 (2H, t, *J* = 13.1 Hz), 4.53–4.62 (3H, m), 7.13–7.57 (15H, m). ^13^C NMR (125 MHz, CDCl_3_): δ = 21.6, 22.2, 25.6, 26.4, 34.9, 37.3, 52.3, 57.0, 59.5, 61.1, 69.5, 72.8, 74.3, 126.8, 127.5, 128.4, 129.2, 129.4, 130.1, 130.3, 132.2, 132.7, 139.1. Found: C, 81.33; H, 8.63; N, 3.04. Anal. Calcd for C_31_H_39_NO_2_: C, 81.36; H, 8.59; N, 3.06.

##### (*S*)-2-((1*R*,2*S*,4*R*)-2-(Benzyloxy)-4-methylcyclohexyl)-1-(dibenzylamino)propan-2-ol (**6b**)

Prepared with **4** with dibenzylamine at reflux for 20 h and eluted by *n*-hexane:EtOAc = 9:1. Yield: 25%, white crystal, m.p = 164–166 °C. ${\left[\alpha \right]}_{D}^{20}$ = −4.0 (c 0.26, MeOH). ^1^H NMR (500 MHz, CDCl_3_): δ = 0.82-0.89 (1H, m), 0.86 (3H, d, *J* = 6.4 Hz), 1.29–1.33 (2H, m), 1.32 (3H, s), 1.56–1.61 (3H, m), 1.73 (1H, dd, *J* = 12.2, 2.1 Hz), 2.10 (1H, dd, *J* = 14.4, 2.4 Hz), 2.63 (1H, d, *J* = 13.2 Hz), 3.48–3.53 (1H, m), 3.60–3.64 (1H, m), 4.04 (1H, s), 4.26 (1H, d, *J* = 11.3 Hz), 4.47–4.66 (5H, m), 7.25–7.65 (15H, m). ^13^C NMR (125 MHz, CDCl_3_): δ = 21.6, 22.2, 26.1, 26.2, 34.8, 37.0, 50.2, 57.1, 58.9, 60.7, 69.9, 73.0, 74.5, 127.8, 127.9, 128.7, 129.2, 129.4, 130.0, 130.1, 132.0, 132.8, 138.4. Found: C, 81.40; H, 8.55; N, 3.07. Anal. Calcd for C_31_H_39_NO_2_: C, 81.36; H, 8.59; N, 3.06.

##### (*S*)-3-(Benzylamino)-2-((1*R*,2*S*,4*R*)-2-(Benzyloxy)-4-methylcyclohexyl)propane-1,2-diol (**12a**)

Prepared with **11a** with benzylamine at reflux for 8 h and eluted by *n*-hexane:EtOAc = 1:2. Yield: 45%, colorless oil. ${\left[\alpha \right]}_{D}^{20}$ = +28.0 (c 0.40, MeOH). ^1^H NMR (500 MHz, CDCl_3_): δ = 0.82–0.89 (1H, m), 0.86 (3H, d, *J* = 6.4 Hz), 1.29–1.33 (2H, m), 1.32 (3H, s), 1.56–1.61 (3H, m), 1.73 (1H, dd, *J* = 12.2, 2.1 Hz), 2.10 (1H, dd, *J* = 14.4, 2.4 Hz), 2.63 (1H, d, *J* = 13.2 Hz), 3.48–3.53 (1H, m), 3.60–3.64 (1H, m), 4.04 (1H, s), 4.26 (1H, d, *J* = 11.3 Hz), 4.47–4.66 (5H, m), 7.25–7.65 (15H, m). ^13^C NMR (125 MHz, CDCl_3_): δ = 21.6, 22.2, 26.1, 26.2, 34.8, 37.0, 50.2, 57.1, 58.9, 60.7, 69.9, 73.0, 74.5, 127.8, 127.9, 128.7, 129.2, 129.4, 130.0, 130.1, 132.0, 132.8, 138.4. Found: C, 81.40; H, 8.55; N, 3.07. Anal. Calcd for C_31_H_39_NO_2_: C, 81.36; H, 8.59; N, 3.06.

##### (*R*)-3-(Benzylamino)-2-((1*R*,2*S*,4*R*)-2-(Benzyloxy)-4-methylcyclohexyl)propane-1,2-diol (**12b**)

Prepared with **11a** with benzylamine at reflux for 8 h and eluted by *n*-hexane:EtOAc = 1:2. Yield: 11%, colorless oil. ${\left[\alpha \right]}_{D}^{20}$ = +19.0 (c 0.30, MeOH). ^1^H NMR (500 MHz, CDCl_3_): δ = 0.86-0.97 (6H, m), 0.88 (3H, d, *J* = 6.4 Hz), 1.25–1.42 (14H, m), 1.57–1.61 (3H, m), 1.70–1.77 (3H, m), 2.12–2.17 (1H, m), 2.76 (2H, s), 3.48 (1H, s), 3.62 (1H, q, *J* = 11.2 Hz), 3.70 (1H, q, *J* = 13.3 Hz), 3.90 (1H, s), 4.23 (1H, d, *J* = 11.0 Hz), 4.57 (1H, d, *J* = 11.0 Hz), 7.22–7.38 (10H, m). ^13^C NMR (125 MHz, CDCl_3_): δ = 21.1, 22.3, 26.1, 34.9, 37.1, 45.1, 54.0, 54.9, 67.3, 70.0, 74.4, 74.8, 127.7, 128.2, 128.3, 128.6, 128.7, 128, 8, 137.8. Found: C, 81.33; H, 8.62; N, 3.11. Anal. Calcd for C_31_H_39_NO_2_: C, 81.36; H, 8.59; N, 3.06.

##### (*S*)-2-((1*R*,2*S*,4*R*)-2-(Benzyloxy)-4-methylcyclohexyl)-3-(dibenzylamino)propane-1,2-diol (**13a**)

Prepared with **11a** with dibenzylamine at reflux for 8 h and eluted by *n*-hexane:EtOAc = 4:1. Yield: 54%, colorless oil. ${\left[\alpha \right]}_{D}^{20}$ = −2.0 (c 0.26, MeOH). ^1^H NMR (500 MHz, CDCl_3_): δ = 0.87 (3H, d, *J* = 6.2 Hz), 0.80–1.00 (4H, m), 1.10–1.25 (1H, m), 1.25–1.35 (2H, m), 1.45–1.80 (7H, m), 2.11 (1H, d, *J* = 14.0 Hz), 3.10 (1H, d, *J* = 13.0 Hz), 3.32 (1H, d, *J* = 8.6 Hz), 3.51 (1H, d, *J* = 12.5 Hz), 3.60 (1H, brs), 3.90–4.10 (1H, m), 4.04 (1H, d, *J* = 12.6 Hz), 4.24 (1H, s), 4.30 (1H, d, *J* = 11.1 H), 4.38 (1H, d, *J* = 11.7 Hz), 4.95 (1H, d, *J* = 11.9 Hz), 5.21 (1H, s), 5.91 (1H, s), 7.25–7.62 (15H, m). ^13^C NMR (125 MHz, CDCl_3_): δ = 21.5, 22.2, 26.1, 34.6, 36.9, 46.4, 57.3, 58.3, 59.5, 66.0, 70.2, 74.8, 74.9, 128.1, 128.2, 128.7, 129.4, 129.6, 130.2, 130.4, 131.5, 138.3. Found: C, 78.63; H, 8.27; N, 3.00. Anal. Calcd for C_31_H_39_NO_3_: C, 78.61; H, 8.30; N, 2.96.

##### (*R*)-2-((1*R*,2*S*,4*R*)-2-(Benzyloxy)-4-methylcyclohexyl)-3-(dibenzylamino)propane-1,2-diol (**13b**)

Prepared with **11b** with dibenzylamine at reflux for 8 h and eluted by *n*-hexane:EtOAc = 4:1. Yield: 7%, colorless oil. ${\left[\alpha \right]}_{D}^{20}$ = +5.0 (c 0.20, MeOH). ^1^H NMR (500 MHz, CDCl_3_): δ = 0.84 (3H, d, *J* = 6.5 Hz), 0.84–0.90 (3H, m), 1.26 (3H, s), 1.25–1.29 (1H, m), 1.48–1.55 (1H, m), 1.57–1.62 (1H, m), 1.70–1.80 (3H, m), 2.05–2.13 (1H, m), 2.69 (2H, m), 3.37 (1H, d, *J* = 11.3 Hz), 3.41 (2H, d, *J* = 13.3 Hz), 3.51 (1H, d, *J* = 11.3 Hz), 3.85 (2H, d, *J* = 13.3 Hz), 4.00 (1H, s), 4.25 (11.1 Hz), 4.55 (1H, d, *J* = 11.2 Hz), 7.23–7.49 (15H, m). ^13^C NMR (125 MHz, CDCl_3_): δ = 20.6, 22.3, 26.2, 34.9, 37.0, 59.2, 60.6, 67.2, 69.9, 75.4, 75.6, 127.5, 127.9, 128.6, 128.7, 129.5, 139.0. Found: C, 78.57; H, 8.33; N, 2.94. Anal. Calcd for C_31_H_39_NO_3_: C, 78.61; H, 8.30; N, 2.96.

##### (*S*)-1-(Benzylamino)-3-(Benzyloxy)-2-((1*R*,2*S*,4*R*)-2-(Benzyloxy)-4-methylcyclohexyl)propan-2-ol (**20a**)

Prepared with **19a** and benzylamine at reflux for 6 h and eluted by *n*-hexane:EtOAc = 2:1. Yield: 77%, colorless oil. ${\left[\alpha \right]}_{D}^{20}$ = +51.0 (c 0.25, MeOH). ^1^H NMR (500 MHz, CDCl_3_): δ = 0.87 (3H, d, *J* = 6.4 Hz), 0.86-0.95 (2H, m), 1.48-1.51 (1H, m), 1.64–1.76 (4H, m), 2.05–2.08 (1H, m), 2.72 (1H, dd, *J* = 16.4, 11.7 Hz), 3.44 (1H, d, *J* = 9.2 Hz), 3.50 (1H, d, *J* = 9.2 Hz), 3.67 (1H, d, *J* = 13.3 Hz), 3.78 (1H, d, *J* = 13.4 Hz), 3.99 (1H, s), 4.13 (1H, d, *J* = 11.2 Hz), 4.42 (1H, d, *J* = 12.0 Hz), 4.49 (1H, d, *J* = 11.2 Hz), 4.50 (1H, d, *J* = 12.0 Hz), 7.22–7.32 (15H, m). ^13^C NMR (125 MHz, CDCl_3_): δ = 20.8, 22.4, 26.2, 35.0, 37.0, 43.8, 54.3, 55.2, 69.8, 73.6, 73.8, 74.9, 75.6, 127.0, 127.7, 127.8, 128.0, 128.4, 128.5, 128.6, 138.4, 138.5. Found: C, 78.59; H, 8.33; N, 2.98. Anal. Calcd for C_31_H_39_NO_3_: C, 78.61; H, 8.30; N, 2.96.

##### (*R*)-1-(Benzylamino)-3-(Benzyloxy)-2-((1*R*,2*S*,4*R*)-2-(Benzyloxy)-4-methylcyclohexyl)propan-2-ol (**20b**)

Prepared with **19b** and benzylamine at reflux for 6 h and eluted by *n*-hexane:EtOAc = 1:2. Yield: 84%, colorless oil. ${\left[\alpha \right]}_{D}^{20}$ = +42.0 (c 0.25, MeOH). ^1^H NMR (500 MHz, CDCl_3_): δ = 0.85–0.97 (3H, m), 0.88 (1H, d, *J* = 6.4 Hz), 1.25–1.29 (1H, m), 1.41–1.46 (1H, m), 1.57–1.65 (2H, m), 1.71–1.77 (2H, m), 2.13–2.17 (1H, m), 2.74 (1H, d, *J* = 12.2 Hz), 2.91 (1H, d, *J* = 12.2 Hz), 3.58 (1H, d, *J* = 11.3 Hz), 3.62 (1H, d, *J* = 11.3 Hz), 3.74 (2H, s), 4.10 (1H, s), 4.33 (1H, d, *J* = 10.9 Hz), 4.60 (1H, d, *J* = 10.9 Hz), 7.20–7.35 (10H, m). ^13^C NMR (125 MHz, CDCl_3_): δ = 21.2, 22.3, 26.1, 34.9, 37.0, 53.5, 53.8, 68.9, 70.2, 74.1, 75.2, 127.9, 128.2, 128.3, 128.8, 137.9. Found: C, 75.12; H, 8.70; N, 2.63. Anal. Calcd for C_24_H_33_NO_3_: C, 75.16; H, 8.67; N, 3.65.

##### (*S*)-1-(Benzyloxy)-2-((1*R*,2*S*,4*R*)-2-(Benzyloxy)-4-methylcyclohexyl)-3-(dibenzylamino)propan-2-ol (**21a**)

Prepared with **19a** and dibenzylamine at reflux for 6 h and eluted by *n*-hexane:EtOAc = 9:1. Yield: 67%, white crystal, m.p. = 54–55 °C. ${\left[\alpha \right]}_{D}^{20}$ = +39.0 (c 0.25, MeOH). ^1^H NMR (500 MHz, CDCl_3_): δ = 0.83 (3H, d, *J* = 6.5 Hz), 0.79–0.90 (2H, m), 1.39–1.41 (1H, m), 1.57–1.69 (4H, m), 1.97–2.00 (1H, m), 2.70 (1H, d, *J* = 13.9 Hz), 2.77 (1H, d, *J* = 13.9 Hz), 3.33 (1H, d, *J* = 9.1 Hz), 3.49 (1H, d, *J* = 13.7 Hz), 3.60 (1H, d, *J* = 9.1 Hz), 3.77 (1H, d, *J* = 13.7 Hz), 3.80 (1H, d, *J* = 9.7 Hz), 3.87 (1H, s), 4.12 (1H, d, *J* = 11.5 Hz), 4.35 (1H, d, *J* = 16.6 Hz), 4.66 (1H, d, *J* = 16.7 Hz), 4.44 (1H, d, *J* = 11.4 Hz), 7.16–7.31 (20H, m). ^13^C NMR (125 MHz, CDCl_3_): δ = 20.9, 22.4, 26.3, 35.2, 37.4, 45.4, 58.2, 60.0, 69.8, 73.1, 73.4, 75.3, 76.0, 126.9, 127.5, 127.6, 127.8, 127.9, 128.2, 128.3, 128.5, 129.3, 138.7, 138.9, 140.0. Found: C, 80.95; H, 8.03; N, 2.50. Anal. Calcd for C_38_H_45_NO_3_: C, 80.96; H, 8.05; N, 2.48.

##### (*R*)-1-(Benzyloxy)-2-((1*R*,2*S*,4*R*)-2-(Benzyloxy)-4-methylcyclohexyl)-3-(dibenzylamino)propan-2-ol (**21b**)

Prepared with **19b** and dibenzylamine at reflux for 6 h and eluted by *n*-hexane:EtOAc = 9:1. Yield: 53%, colorless oil. ${\left[\alpha \right]}_{D}^{20}$ = +26.0 (c 0.25, MeOH). ^1^H NMR (500 MHz, CDCl_3_): δ = 0.84 (3H, d, *J* = 6.5 Hz), 0.79–0.83 (3H, m), 1.51–1.58 (4H, m), 1.67–1.76 (3H, m), 1.97–2.01 (1H, m), 2.60 (1H, d, *J* = 13.7 Hz), 2.71 (1H, d, *J* = 13.6 Hz), 3.23 (1H, d, *J* = 8.7 Hz), 3.40 (2H, d, *J* = 13.9 Hz), 3.65 (1H, s), 3.73 (1H, d, *J* = 8.7 Hz), 3.80 (1H, s), 3.88 (2H, d, *J* = 13.9 Hz), 3.96 (1H, d, *J* = 12.7 Hz), 4.17 (1H, d, *J* = 11.9 Hz), 4.34 (1H, d, *J* = 12.0 Hz), 4.40 (1H, d, *J* = 11.3 Hz), 7.17–7.32 (20H, m). ^13^C NMR (125 MHz, CDCl_3_): δ = 20.8, 22.4, 26.2, 35.0, 36.9, 44.3, 57.5, 59.8, 69.5, 72.5, 73.2, 75.9, 76.5, 126.7, 127.6, 127.7, 127.8, 128.1, 128.2, 128.3, 128.5, 129.2, 138.5, 138.6, 140.2. Found: C, 81.00; H, 8.10; N, 2.45. Anal. Calcd for C_38_H_45_NO_3_: C, 80.96; H, 8.05; N, 2.48.

##### (1*S*,2*R*,5*R*)-2-((*S*)-1-(Benzylamino)-3-(Benzyloxy)-2-hydroxypropan-2-yl)-5-methylcyclohexanol (**25a**)

Prepared with **25a** and benzylamine at reflux for 8 h and eluted by *n*-hexane:EtOAc = 1:2. Yield: 71%, colorless oil. ${\left[\alpha \right]}_{D}^{20}$ = +18.0 (c 0.29, MeOH). ^1^H NMR (500 MHz, CDCl_3_): δ = 0.85 (3H, d, *J* = 6.4 Hz), 0.84–0.92 (2H, m), 1.00–1.06 (1H, m), 1.20–1.29 (2H, m), 1.53–1.62 (2H, m), 1.70–1.73 (1H, m), 1.86–1.94 (2H, m), 2.64 (1H, d, *J* = 11.9 Hz), 2.75 (1H, d, *J* = 12.0 Hz), 3.33 (1H, d, *J* = 9.2 Hz), 3.38 (1H, d, *J* = 9.2 Hz), 3.80 (2H, s), 4.23 (1H, s), 4.51 (1H, d, *J* = 16.8 Hz), 4.52 (1H, d, *J* = 16.8 Hz), 7.25–7.35 (10H, m). ^13^C NMR (125 MHz, CDCl_3_): δ = 21.5, 22.4, 26.0, 35.5, 42.4, 48.2, 50.6, 53.8, 64.6, 73.8, 74.7, 75.1, 127.7, 127.9, 128.1, 128.6, 128.7, 128.8, 137.7, 138.1. Found: C, 75.13; H, 8.65; N, 3.70. Anal. Calcd for C_24_H_33_NO_3_: C, 75.16; H, 8.67; N, 3.65.

##### (1*S*,2*R*,5*R*)-2-((*R*)-1-(Benzylamino)-3-(Benzyloxy)-2-hydroxypropan-2-yl)-5-methylcyclohexanol (**25b**)

Prepared with **25b** and benzylamine at reflux for 8 h and eluted by *n*-hexane:EtOAc = 1:2. Yield: 85%, colorless oil. ${\left[\alpha \right]}_{D}^{20}$ = +4.0 (c 0.25, MeOH). ^1^H NMR (500 MHz, CDCl_3_): δ = 0.85 (3H, d, *J* = 6.4 Hz), 0.84–0.90 (3H, m), 0.98–1.03 (1H, m), 1.25–1.42 (7H, m), 1.48–1.56 (2H, m), 1.63–1.77 (2H, m), 1.82–1.90 (2H, m), 2.80 (2H, s), 3.37 (1H, d, *J* = 9.3 Hz), 3.53 (1H, d, *J* = 9.3 Hz), 3.77 (2H, s), 4.07 (1H, s), 4.47 (1H, d, *J* = 11.9 Hz), 4.53 (1H, d, *J* = 11.9 Hz), 7.24–7.34 (10H, m). ^13^C NMR (125 MHz, CDCl_3_): δ = 20.4, 22.4, 26.0, 29.8, 35.3, 42.3, 46.8, 52.1, 54.2, 66.2, 73.4, 73.6, 75.0, 127.6, 127.9, 128.1, 128.5, 128.7, 128.8, 137.8, 138.6. Found: C, 75.20; H, 8.70; N, 3.63. Anal. Calcd for C_24_H_33_NO_3_: C, 75.16; H, 8.67; N, 3.65.

##### (1*S*,2*R*,5*R*)-2-((*S*)-1-(Benzyloxy)-3-(dibenzylamino)-2-hydroxypropan-2-yl)-5-methylcyclohexanol (**26a**)

Prepared with **25a** and dibenzylamine at reflux for 8 h and eluted by *n*-hexane:EtOAc = 9:1. Yield: 88%, colorless oil. ${\left[\alpha \right]}_{D}^{20}$ = +23.0 (c 0.25, MeOH). ^1^H NMR (500 MHz, CDCl_3_): δ = 0.82 (3H, d, *J* = 6.4 Hz), 0.76–0.89 (2H, m), 0.92–0.98 (1H, m), 1.19–1.45 (3H, m), 1.60–1.64 (1H, m), 1.77–1.83 (2H, m), 2.67 (1H, d, *J* = 14.0 Hz), 2.79 (1H, d, *J* = 14.0 Hz), 3.25 (1H, d, *J* = 9.0 Hz), 3.42 (1H, d, *J* = 9.0 Hz), 3.53 (2H, d, *J* = 13.5 Hz), 3.80 (2H, d, *J* = 13.5 Hz), 4.16 (1H, s), 4.33 (1H, d, *J* = 11.7 Hz), 4.39 (1H, d, *J* = 11.7 Hz), 7.21–7.33 (15H, m). ^13^C NMR (125 MHz, CDCl_3_): δ = 20.2, 22.4, 25.9, 35.3, 42.1, 46.1, 56.1, 60.3, 66.2, 72.5, 73.5, 74.8, 127.5, 128.0, 128.5, 128.6, 129.5, 137.6, 138.6. Found: C, 78.63; H, 8.33; N, 2.98. Anal. Calcd for C_31_H_39_NO_3_: C, 78.61; H, 8.30; N, 2.96.

##### (1*S*,2*R*,5*R*)-2-((*R*)-1-(Benzyloxy)-3-(dibenzylamino)-2-hydroxypropan-2-yl)-5-methylcyclohexanol (**26b**)

Prepared with **25b** and dibenzylamine at reflux for 8 h and eluted by *n*-hexane:EtOAc = 9:1. Yield: 88%, colorless oil. ${\left[\alpha \right]}_{D}^{20}$ = +8.0 (c 0.25, MeOH). ^1^H NMR (500 MHz, CDCl_3_): δ = 0.82 (3H, d, *J* = 6.3 Hz), 0.77–0.85 (1H, m), 0.88–0.94 (1H, m), 1.29–1.33 (1H, m), 1.39–1.42 (1H, m), 1.49–1.55 (3H, m), 1.64–1.67 (1H, m), 1.77–1.80 (2H, m), 2.82 (1H, d, *J* = 17.0 Hz), 2.83 (1H, d, *J* = 16.9 Hz), 3.16 (1H, brs), 3.18 (1H, d, *J* = 9.1 Hz), 3.34 (1H, d, *J* = 9.1 Hz), 3.61 (2H, d, *J* = 13.5 Hz), 3.72 (2H, d, *J* = 13.5 Hz), 3.76 (1H, brs), 4.06 (1H, s), 4.36 (2H, s), 7.22–7.33 (15H, m). ^13^C NMR (125 MHz, CDCl_3_): δ = 20.6, 22.4, 25.8, 35.4, 42.2, 44.6, 57.0, 60.6, 66.7, 71.7, 73.5, 75.4, 127.5, 128.0, 128.5, 128.7, 129.4, 137.8, 139.1. Found: C, 78.57; H, 8.35; N, 2.93. Anal. Calcd for C_31_H_39_NO_3_: C, 78.61; H, 8.30; N, 2.96.

##### (*S*)-2-((1*R*,2*R*,4*R*)-2-(Benzyloxy)-4-methylcyclohexyl)-1-(dibenzylamino)propan-2-ol (**30b**)

Prepared with **29** and dibenzylamine at reflux for 20 h and eluted by *n*-hexane:EtOAc = 9:1. Yield: 47%, colorless oil. ${\left[\alpha \right]}_{D}^{20}$ = −40.0 (c 0.255, MeOH). ^1^H NMR (500 MHz, CDCl_3_): δ = 0.45–0.53 (1H, m), 0.84–0.90 (1H, m), 0.93 (3H, s), 0.96 (3H, d, *J* = 5.5 Hz), 1.04 (1H, q, *J* = 12.0 Hz), 1.23–1.43 (3H, m), 1.55 (3H, s), 2.03–2.08 (1H, m), 2.22–2.26 (1H, m), 2.31 (1H, d, *J* = 13.6 Hz), 2.45 (1H, d, *J* = 13.7 Hz), 3.23 (2H, d, *J* = 13.7 hz), 3.54 (1H, td, *J* = 10.6, 3.9 Hz), 4.18 (2H, d, *J* = 13.6 Hz), 4.39 (1H, d, *J* = 11.0 Hz), 4.66 (1H, d, *J* = 11.0 Hz), 5.25 (1H, s), 7.18–7.35 (15H, m). ^13^C NMR (125 MHz, CDCl_3_): δ = 22.3, 23.6, 26.6, 31.8, 35.8, 40.0, 47.9, 59.8, 61.4, 70.2, 76.8, 81.2, 126.7, 128.1, 128.3, 128.7, 129.5, 137.6, 140.4. Found: C, 81.37; H, 8.35; N, 2.93. Anal. Calcd for C_31_H_39_NO_2_: C, 81.36; H, 8.33; N, 2.94.

##### (*S*)-2-((1*R*,2*R*,4*R*)-2-(Benzyloxy)-4-methylcyclohexyl)-3-(dibenzylamino)propane-1,2-diol (**34a**)

Prepared with **33** and dibenzylamine at reflux for 8 h and eluted by *n*-hexane:EtOAc = 9:1. Yield: 76%, colorless oil. ${\left[\alpha \right]}_{D}^{20}$ = −126.0 (c 0.30, MeOH). ^1^H NMR (500 MHz, CDCl_3_): δ = 0.55–0.64 (1H, m), 0.77–0.86 (1H, m), 0.87–0.94 (1H, m), 0.90 (3H, d, *J* = 6.5 Hz), 1.22–1.32 (1H, m), 1.51–1.61 (5H, m), 2.22 (1H, d, *J* = 12.1 Hz), 2.43 (1H, d, *J* = 13.6 Hz), 3.13 (2H, d, *J* = 13.3 Hz), 3.38 (1H, d, *J* = 11.5 Hz), 3.46 (1H td, *J* = 10.5, 3.95 Hz), 3.60 (1H, d, *J* = 11.4 Hz), 4.14 (2H, d, *J* = 13.3 Hz), 4.39 (1H, d, *J* = 11.1 Hz), 4.68 (1H, d, *J* = 11.1 Hz), 7.21–7.38 (15H, m). ^13^C NMR (125 MHz, CDCl_3_): δ = 22.2, 26.0, 31.2, 34.4, 40.0, 49.8, 57.2, 60.6, 67.8, 70.1, 76.7, 80.0, 127.2, 128.2, 128.3, 128.5, 128.8, 129.3, 137.7, 139.2. Found: C, 78.58; H, 8.33; N, 2.94. Anal. Calcd for C_31_H_39_NO_3_: C, 78.61; H, 8.30; N, 2.96.

##### (*S*)-1-(Benzyloxy)-2-((1*R*,2*R*,4*R*)-2-(Benzyloxy)-4-methylcyclohexyl)-3-(dibenzylamino)propan-2-ol (**38a**)

Prepared with **37a** and dibenzylamine at reflux for 6 h and eluted by *n*-hexane:EtOAc = 9:1. Yield: 80%, colorless oil. ${\left[\alpha \right]}_{D}^{20}$ = −68.0 (c 0.27, MeOH). ^1^H NMR (500 MHz, CDCl_3_): δ = 0.52–0.60 (1H, m), 0.79–0.87 (1H, m), 0.89 (3H, d, *J* = 6.5 Hz), 0.89–0.96 (1H, m), 1.20–1.35 (H, m), 1.45–1.55 (3H, m), 1.86 (1H, td, *J* = 12.2, 3.2 Hz), 2.16 (1H, d, *J* = 12.2 Hz), 2.43 (1H, d, *J* = 13.6 Hz), 2.61 (1H, d, *J* = 13.6 Hz), 3.13 (1H, d, *J* = 10.6 Hz), 3.19 (2H, d, *J* = 13.6 Hz), 3.45 (1H, td, *J* = 10.6, 3.9 Hz), 3.71 (1H, d, *J* = 10.6 Hz), 4.04 (2H, d, *J* = 13.6 Hz), 4.34 (1H, d, *J* = 11.2 Hz), 4.42 (1H, d, *J* = 12.1 Hz), 4.63 (1H, d, *J* = 12.2 Hz), 4.64 (1H, d, *J* = 11.2 Hz), 4.77 (1H, brs), 7.16–7.34 (20H, m). ^13^C NMR (125 MHz, CDCl_3_): δ = 22.2, 26.5, 31.5, 34.6, 40.2, 48.6, 57.7, 60.2, 70.0, 73.8, 74.3, 77.7, 80.1, 126.8, 127.4, 127.9, 128.1, 128.2, 128.3, 128.6, 129.2, 138.1, 139.0, 140.0. Found: C, 80.93; H, 8.07; N, 2.50. Anal. Calcd for C_38_H_45_NO_3_: C, 80.96; H, 8.05; N, 2.48.

##### (*R*)-1-(Benzyloxy)-2-((1*R*,2*R*,4*R*)-2-(Benzyloxy)-4-methylcyclohexyl)-3-(dibenzylamino)propan-2-ol (**38b**)

Prepared with **37a** and dibenzylamine at reflux for 6 h and eluted by *n*-hexane:EtOAc = 9:1. Yield: 67%, colorless oil. ${\left[\alpha \right]}_{D}^{20}$ = −37.0 (c 0.25, MeOH). ^1^H NMR (500 MHz, CDCl_3_): δ = 0.82–0.99 (3H, m), 0.93 (3H, d, *J* = 6.5 Hz), 1.29–1.35 (1H, m), 1.44–1.50 (1H, m), 1.55 (1H, brs), 2.00–2.04 (1H, m), 2.17 (1H, dd, *J* = 12.1, 1.4 Hz), 2.39 (1H, d, *J* = 13.7 Hz), 2.62 (1H, d, *J* = 13.7 Hz), 3.29 (2H, s), 3.33 (2H, d, *J* = 13.7 Hz), 3.68 (1H, td, *J* = 10.7, 3.9 Hz), 4.08 (2H, d, *J* = 13.7 Hz), 4.19 (1H, d, *J* = 10.9 Hz), 4.41 (2H, q, *J* = 12.1 Hz), 4.51 (1H, d, *J* = 10.9 Hz), 5.35 (1H, brs), 7.18–7.33 (20H, m). ^13^C NMR (125 MHz, CDCl_3_): δ = 22.3, 26.3, 31.7, 35.1, 40.5, 47.7, 58.0, 59.9, 70.1, 73.7, 74.8, 77.9, 81.6, 126.7, 127.6, 127.9, 128.1, 128.4, 128.6, 129.5, 138.0, 138.7, 140.2. Found: C, 80.95; H, 8.07; N, 2.52. Anal. Calcd for C_38_H_45_NO_3_: C, 80.96; H, 8.05; N, 2.48.

##### (1*R*,2*R*,5*R*)-2-((*S*)-1-(Benzyloxy)-3-(dibenzylamino)-2-hydroxypropan-2-yl)-5-methylcyclohexanol (**41a**)

Prepared with **37b** and dibenzylamine at reflux for 8 h and eluted by *n*-hexane:EtOAc = 9:1. Yield: 88%, colorless oil. ${\left[\alpha \right]}_{D}^{20}$ = −5.0 (c 0.285, MeOH). ^1^H NMR (500 MHz, CDCl_3_): δ = 0.60–0.69 (1H, m), 0.74–0.83 (1H, m), 0.85–0.95 (1H, m), 0.86 (3H, d, *J* = 6.5 Hz), 1.22–1.32 (1H, m), 1.45–1.55 (2H, m), 1.55 (1H, s), 1.64–1.69 (1H, m), 1.86–1.89 (1H, m), 2.62 (1H, d, *J* = 14.1 Hz), 2.83 (1H, d, *J* = 14.1 Hz), 3.16 (1H, d, *J* = 9.9 Hz), 3.41–3.47 (2H, m), 3.58 (2H, d, *J* = 13.4 Hz), 3.75 (2H, d, *J* = 13.4 Hz), 3.88 (1H, brs), 4.41 (1H, d, *J* = 12.0 Hz), 4.52 (1H, d, *J* = 12.0 Hz), 4.77 (1H, brs), 7.24–7.32 (15H, m). ^13^C NMR (125 MHz, CDCl_3_): δ = 22.1, 25.6, 31.2, 34.7, 44.4, 49.3, 54.9, 60.4, 71.0, 73.6, 74.8, 77.5, 127.4, 127.8, 128.0, 128.5, 128.6, 129.4, 138.2, 139.1. Found: C, 78.58; H, 8.27; N, 2.95. Anal. Calcd for C_31_H_39_NO_3_: C, 78.61; H, 8.30; N, 2.96.

##### (1*R*,2*R*,5*R*)-2-((*R*)-1-(Benzyloxy)-3-(dibenzylamino)-2-hydroxypropan-2-yl)-5-methylcyclohexanol (**41b**)

Prepared with **37b** and dibenzylamine at reflux for 8 h and eluted by *n*-hexane:EtOAc = 9:1. Yield: 76%, colorless oil. ${\left[\alpha \right]}_{D}^{20}$ = −22.0 (c 0.28, MeOH). ^1^H NMR (500 MHz, CDCl_3_): δ = 0.64–0.72 (1H, m), 0.87 (3H, d, *J* = 6.4 Hz), 0.86–0.95 (2H, m), 1.26–1.49 (5H, m), 1.59 (1H, brs), 1.92 (1H, d, *J* = 12.4 Hz), 2.62 (1H, d, *J* = 13.9 Hz), 2.87 (1H, d, *J* = 13.9 Hz), 3.39 (2H, s), 3.51 (1H, d, *J* = 13.4 Hz), 3.70 (1H, td, *J* = 10.3, 4.3 Hz), 3.87 (1H, d, *J* = 13.4 Hz), 4.43 (2H, t, *J* = 12.3 Hz), 7.24–7.32 (15H, m). ^13^C NMR (125 MHz, CDCl_3_): δ = 22.1, 24.5, 31.3, 34.9, 44.7, 50.5, 58.1, 60.1, 71.8, 73.2, 73.7, 76.3, 127.4, 127.8, 128.5, 128.6, 129.4, 138.1, 139.0. Found: C, 78.64; H, 8.33; N, 2.99. Anal. Calcd for C_31_H_39_NO_3_: C, 78.61; H, 8.30; N, 2.96.

#### 4.2.6. General Procedure for Ring-Opening of Epoxide with Azoles

A solution of epoxides (2.9 mmol) in dry DMF (30 mL) was added to the triazole or imidazole (8.7 mmol) in dry DMF (10 mL) and K_2_CO_3_ (14.5 mmol). The mixture was kept at reflux temperature for 12–96 h. When the reaction completed (indicated by TLC), the mixture was dissolved in water (15 mL) and extracted with EtOAc (3 × 50 mL). The combined organic phase was again extracted with saturated NaCl solution (3 × 50 mL) then dried (Na_2_SO_4_), filtered, and concentrated. The crude product was purified by column chromatography on silica gel with CHCl_3_:MeOH = 19:1, resulting in *O*-benzyl derivatives, respectively.

##### (*R*)-2-((1R,2S,4R)-2-(Benzyloxy)-4-methylcyclohexyl)-1-(1H-imidazol-1-yl)propan-2-ol (**7a**)

Prepared with **4a** and imidazole at reflux for 24 h. Yield: 42%, colorless oil. ${\left[\alpha \right]}_{D}^{20}$ = +27.0 (c 0.27, MeOH). ^1^H NMR (500 MHz, CDCl_3_): δ = 0.86–0.97 (3H, m), 0.91 (3H, d, *J* = 6.5 Hz), 0.94 (3H, s), 1.23–1.42 (7H, m), 1.68–1.85 (4H, m), 2.22 (1H, dd, *J* = 14.4, 2.2 Hz), 3.82 (2H, d, *J* = 2.8 Hz), 4.07 (1H, s), 4.33 (1H, d, *J* = 11.4 Hz), 4.70 (1H, d, *J* = 11.4 Hz), 6.85 (1H, s), 7.00 (1H, s), 7.25–7.39 (6H, m). ^13^C NMR (125 MHz, CDCl_3_): δ = 21.4, 22.2, 23.3, 26.0, 29.8, 34.8, 36.9, 47.3, 56.2, 69.8, 74.0, 74.6, 120.7, 128.4, 128.5, 128.8, 128.9, 137.6, 138.5. Found: C, 73.10; H, 8.57; N, 8.55. Anal. Calcd for C_20_H_28_N_2_O_2_: C, 73.14; H, 8.59; N, 8.53.

##### (*S*)-2-((1*R*,2*S*,4*R*)-2-(Benzyloxy)-4-methylcyclohexyl)-1-(1H-imidazol-1-yl)propan-2-ol (**7b**)

Prepared with **4b** and imidazole at reflux for 24 h. Yield: 67%, colorless oil. ${\left[\alpha \right]}_{D}^{20}$ = +30.0 (c 0.26, MeOH). ^1^H NMR (500 MHz, CDCl_3_): δ = 0.86–0.94 (1H, m), 0.91 (3H, d, *J* = 6.6 Hz), 0.96–1.02 (1H, m), 1.07 (3H, s), 1.25–1.31 (2H, m), 1.69–1.98 (6H, m), 2.16–2.22 (1H, m), 3.82 (1H, d, *J* = 14.1 Hz), 3.96 (1H, d, *J* = 14.1 Hz), 4.06 (1H, d, *J* = 1.7 Hz), 4.33 (1H, d, *J* = 11.1 Hz), 4.65 (1H, d, *J* = 11.1 Hz), 6.90 (1H, s), 7.02 (1H, s), 7.25–7.26 (5H, m), 7.43 (1H, s). ^13^C NMR (125 MHz, CDCl_3_): δ = 21.3, 22.2, 24.7, 26.0, 34.6, 37.0, 46.4, 55.1, 70.0, 73.8, 75.7, 120.7, 128.1, 128.2, 128.8, 129.0, 137.8, 138.5. Found: C, 73.17; H, 8.62; N, 8.50. Anal. Calcd for C_20_H_28_N_2_O_2_: C, 73.14; H, 8.59; N, 8.53.

##### (*R*)-2-((1*R*,2*S*,4*R*)-2-(Benzyloxy)-4-methylcyclohexyl)-1-(1H-1,2,4-triazol-1-yl)propan-2-ol (**8a**)

Prepared with **4a** and 1,2,4-triazole at reflux for 24 h. Yield: 67%, colorless oil. ${\left[\alpha \right]}_{D}^{20}$ = +42.0 (c 0.275, MeOH). ^1^H NMR (500 MHz, CDCl_3_): δ = 0.91 (3H, d, *J* = 6.5 Hz), 0.94 (3H, s), 0.90–0.98 (2H, m), 1.39–1.43 (1H, m), 1.61 (1H, brs), 1.71–1.84 (4H, m), 2.19–2.23 (1H, m), 4.00 (1H, d, *J* = 13.9 Hz), 4.21 (3H, t, *J* = 14.0 Hz), 4.34 (1H, d, *J* = 11.3 Hz), 4.71 (1H, d, *J* = 11.3 Hz), 7.26 (1H, s), 7.31–7.40 (5H, m), 7.89 (1H, d, *J* = 2.8 Hz). ^13^C NMR (125 MHz, CDCl_3_): δ = 21.2, 22.3, 23.6, 26.0, 34.8, 36.9, 47.0, 57.9, 69.9, 74.0, 75.0, 128.4, 128.9, 137.9, 144.4, 151.6. Found: C, 69.30; H, 8.24; N, 12.80. Anal. Calcd for C_20_H_28_N_2_O_2_: C_19_H_27_N_3_O_2_: C, 69.27; H, 8.26; N, 12.76.

##### (*S*)-2-((1*R*,2*S*,4*R*)-2-(Benzyloxy)-4-methylcyclohexyl)-1-(1H-1,2,4-triazol-1-yl)propan-2-ol (**8b**)

Prepared with **4b** and 1,2,4-triazole at reflux for 24 h. Yield: 67%, colorless oil. ${\left[\alpha \right]}_{D}^{20}$ = +73.0 (c 0.28, MeOH). ^1^H NMR (500 MHz, CDCl_3_): δ = 0.86–1.03 (2H, m), 0.91 (3H, d, *J* = 6.5 Hz), 1.06 (3H, s), 1.37–1.41 (1H, m), 1.56 (1H, s), 1.71–1.90 (4H, m), 2.16–2.21 (1H, m), 3.86 (1H, s), 4.10 (1H, d, *J* = 1.8 Hz), 4.11 (1H, d, *J* = 13.9 Hz), 4.21 (1H, d, *J* = 14.0 Hz), 4.34 (1H, d, *J* = 11.2 Hz), 4.64 (1H, d, *J* = 11.2 Hz), 7.25–7.35 (5H, m), 7.89 (1H, s), 8.06 (1H, s). ^13^C NMR (125 MHz, CDCl_3_): δ = 21.4, 22.3, 24.7, 26.0, 47.0, 57.2, 70.0, 73.7, 75.4, 128.1, 128.7, 137.9, 144.6, 151.5. Found: C, 69.24; H, 8.30; N, 12.73. Anal. Calcd for C_20_H_28_N_2_O_2_: C_19_H_27_N_3_O_2_: C, 69.27; H, 8.26; N, 12.76.

##### (*S*)-2-((1*R*,2*S*,4*R*)-2-(Benzyloxy)-4-methylcyclohexyl)-3-(1H-imidazol-1-yl)propane-1,2-diol (**14a**)

Prepared with **11a** and 1,2,4-triazole at reflux for 12 h. Yield: 58%, colorless oil. ${\left[\alpha \right]}_{D}^{20}$ = +44.0 (c 0.25, MeOH). ^1^H NMR (500 MHz, CDCl_3_): δ = 0.91 (3H, d, *J* = 6.6 Hz), 0.94–1.03 (2H, m), 1.65–1.71 (2H, m), 1.77–1.80 (1H, m), 1.85–1.95 (2H, m), 2.10–2.20 (1H, m), 3.05 (1H, d, *J* = 10.9 Hz), 3.32 (1H, d, *J* = 10.9 Hz), 3.45 (1H, s), 4.00 (1H, d, *J* = 14.0 Hz), 4.06 (1H, d, *J* = 14.0 Hz), 4.33 (1H, d, *J* = 11.1 Hz), 4.63 (1H, d, *J* = 11.1 Hz), 6.92 (1H, s), 6.96 (1H, s), 7.25–7.33 (5H, m), 7.46 (1H, s). ^13^C NMR (125 MHz, CDCl_3_): δ = 20.8, 22.3, 26.0, 34.7, 36.8, 42.7, 50.7, 62.4, 69.9, 75.6, 75.8, 120.9, 128.1, 128.7, 137.7, 138.7. Found: C, 69.77; H, 8.16; N, 8.10. Anal. Calcd for C_20_H_28_N_2_O_3_: C, 69.74; H, 8.19; N, 8.13.

##### (*S*)-2-((1*R*,2*S*,4*R*)-2-(Benzyloxy)-4-methylcyclohexyl)-3-(1H-1,2,4-triazol-1-yl)propane-1,2-diol (**15a**)

Prepared with **11a** and 1,2,4-triazole at reflux for 12 h. Yield: 46%, colorless oil. ${\left[\alpha \right]}_{D}^{20}$ = +50.0 (c 0.26, MeOH). ^1^H NMR (500 MHz, CDCl_3_): δ = 0.86–0.93 (1H, m), 0.91 (3H, d, *J* = 6.6 Hz), 0.95–1.05 (2H, m), 1.25–1.29 (1H, m), 1.55 (2H, s), 1.65–1.70 (1H, m), 1.73–1.85 (1H, m), 1.84–1.93 (2H, m), 2.15–2.23 (1H, m), 2.99 (1H, t, *J* = 7.6 Hz), 3.07 (1H, dd, *J* = 12.0, 4.4 Hz), 3.36 (1H, dd, *J* = 11.9, 7.9 Hz), 4.12 (2H, s), 4.28 (1H, d, *J* = 6.9 Hz), 4.32 (1H, d, *J* = 11.2 Hz), 4.64 (1H, d, *J* = 11.1 Hz), 7.25–7.34 (5H, m), 7.92 (1H, s), 8.05 (1H, s). ^13^C NMR (125 MHz, CDCl_3_): δ = 20.8, 22.2, 26.0, 34.6, 36.7, 42.5, 53.7, 64.3, 69.9, 75.2, 75.9, 128.2, 128.8, 137.5, 151.9. Found: C, 66.10; H, 7.83; N, 12.11. Anal. Calcd for C_19_H_27_N_3_O_3_: C, 66.06; H, 7.88; N, 12.16.

##### (*S*)-1-(Benzyloxy)-2-((1*R*,2*S*,4*R*)-2-(Benzyloxy)-4-methylcyclohexyl)-3-(1H-imidazol-1-yl)propan-2-ol (**22a**)

Prepared with **19a** and imidazole at reflux for 48 h. Yield: 50%, colorless oil. ${\left[\alpha \right]}_{D}^{20}$ = +47.0 (c 0.20, MeOH). ^1^H NMR (500 MHz, CDCl_3_): δ = 0.92 (3H, d, *J* = 6.4 Hz), 0.91–0.97 (2H, m), 1.52–1.81 (8H, m), 2.21 (1H, dd, *J* = 14.3, 2.4 Hz), 2.65 (1H, d, *J* = 9.3 Hz), 3.17 (1H, d, *J* = 9.3 Hz), 3.81 (1H, d, *J* = 13.8 Hz), 4.02 (1H, d, *J* = 13.8 Hz), 4.02 (1H, brs), 4.11 (1H, brs), 4.27 (1H, d, *J* = 11.7 Hz), 4.31 (1H, d, *J* = 11.4 Hz), 4.40 (1H, d, *J* = 11.7 Hz), 4.69 (1H, d, *J* = 11.3 Hz), 6.84 (1H, s), 6.98 (1H, s), 7.25–7.40 (11H, m). ^13^C NMR (125 MHz, CDCl_3_): δ = 20.7, 22.3, 25.9, 34.6, 36.8, 42.7, 52.5, 69.1, 69.8, 73.2, 75.0, 75.4, 120.8, 127.9, 128.5, 128.6, 128.9, 137.4, 137.9, 138.6. Found: C, 74.63; H, 7.93; N, 6.47. Anal. Calcd for C_27_H_34_N_2_O_3_: C, 74.62; H, 7.89; N, 6.45.

##### (*R*)-1-(Benzyloxy)-2-((1*R*,2*S*,4*R*)-2-(Benzyloxy)-4-methylcyclohexyl)-3-(1H-imidazol-1-yl)propan-2-ol (**22b**)

Prepared with **19b** and imidazole at reflux for 48 h. Yield: 42%, colorless oil. ${\left[\alpha \right]}_{D}^{20}$ = +71.0 (c 0.20, MeOH). ^1^H NMR (500 MHz, CDCl_3_): δ = 0.87–0.92 (1H, m), 0.95–1.03 (1H, m), 1.57–1.68 (6H, m), 1.73–1.76 (1H, m), 1.82–1.87 (1H, m), 1.90–1.94 (1H, m), 2.09–1.13 (1H, m), 2.74 (1H, d, *J* = 9.4 Hz), 3.06 (1H, d, *J* = 9.4 Hz), 3.82 (1H, s), 3.93 (1H, d, *J* = 14.0 Hz), 3.95 (1H, d, *J* = 14.1 Hz), 4.04 (1H, d, *J* = 11.1 Hz), 4.08 (1H, d, *J* = 13.9 Hz), 4.19 (1H, d, *J* = 11.7 Hz), 4.53 (1H, d, *J* = 11.7 Hz), 4.50 (1H, d, *J* = 11.1 Hz), 6.90 (1H, s), 6.99 (1H, s), 7.19–7.42 (11H, m). ^13^C NMR (125 MHz, CDCl_3_): δ = 20.6, 22.3, 25.9, 34.5, 36.6, 42.4, 51.6, 69.7, 69.9, 73.2, 75.0, 75.5, 120.9, 128.2, 128.3, 128.6, 128.7, 137.5. Found: C, 74.59; H, 7.87; N, 6.43. Anal. Calcd for C_27_H_34_N_2_O_3_: C, 74.62; H, 7.89; N, 6.45.

##### (*S*)-1-(Benzyloxy)-2-((1*R*,2*S*,4*R*)-2-(Benzyloxy)-4-methylcyclohexyl)-3-(1H-1,2,4-triazol-1-yl)propan-2-ol (**23a**)

Prepared with **19a** and 1,2,4-triazole at reflux for 48 h. Yield: 67%, colorless oil. ${\left[\alpha \right]}_{D}^{20}$ = +52.0 (c 0.25, MeOH). ^1^H NMR (500 MHz, CDCl_3_): δ = 0.89–0.98 (2H, m), 0.90 (3H, d, *J* = 6.4 Hz), 1.54–1.65 (3H, m), 1.71–7.79 (3H, m), 2.18 (1H, dd, *J* = 14.4, 2.2 Hz), 3.05 (1H, d, *J* = 9.7 Hz), 3.15 (1H, d, *J* = 9.7 Hz), 4.22 (2H, d, *J* = 14.1 Hz), 4.31–4.39 (4H, m), 4.46 (1H, d, *J* = 11.7 Hz), 4.65 (1H, d, *J* = 11.2 Hz), 7.25–7.36 (10H, m), 7.88 (1H, s), 7.94 (1H, s). ^13^C NMR (125 MHz, CDCl_3_): δ = 20.8, 22.3, 26.0, 34.7, 36.8, 43.4, 53.7, 69.9, 71.3, 73.6, 75.1, 75.6, 128.0, 128.1, 128.2, 128.3, 128.5, 128.8, 137.8, 137.9, 144.8, 151.4. Found: C, 71.67; H, 7.69; N, 9.63. Anal. Calcd for C_26_H_33_N_3_O_3_: C, 71.70; H, 7.64; N, 9.65.

##### (*R*)-1-(Benzyloxy)-2-((1*R*,2*S*,4*R*)-2-(benzyloxy)-4-methylcyclohexyl)-3-(1H-1,2,4-triazol-1-yl)propan-2-ol (**23b**)

Prepared with **19b** and 1,2,4-triazole at reflux for 48 h. Yield: 67%, colorless oil. ${\left[\alpha \right]}_{D}^{20}$ = +60.0 (c 0.25, MeOH). ^1^H NMR (500 MHz, CDCl_3_): δ = 0.88–0.94 (2H, m), 0.90 (3H, d, *J* = 6.6 Hz), 0.98–1.02 (1H, m), 1.56 (3H, s), 1.63–1.89 (5H, m), 2.09–2.13 (1H, m), 2.94 (1H, d, *J =* 9.6 Hz), 3.11 (1H, d, *J* = 9.6 Hz), 3.90 (1H, s), 4.00 (1H, s), 4.09 (1H, d, *J* = 11.1 Hz), 4.27 (1H, d, *J* = 11.7 Hz), 4.28 (1H, d, *J* = 15.9 Hz), 4.35 (1H, d, *J* = 14.0 Hz), 4.49 (1H, d, *J* = 11.8 Hz), 4.52 (1H, d, *J* = 11.1 Hz), 7.21–7.37 (10H, m), 7.89 (1H, s), 8.04 (1H, s). ^13^C NMR (125 MHz, CDCl_3_): δ = 20.7, 22.3, 26.0, 34.6, 36.7, 42.6, 53.4, 69.8, 70.8, 73.4, 74.9, 75.3, 128.0, 128.1, 128.2, 128.3, 128.5, 128.7, 137.7, 144.9, 151.3. Found: C, 71.73; H, 7.60; N, 9.68. Anal. Calcd for C_26_H_33_N_3_O_3_: C, 71.70; H, 7.64; N, 9.65.

##### (1*S*,2*R*,5*R*)-2-((*S*)-1-(Benzyloxy)-2-hydroxy-3-(1H-imidazol-1-yl)propan-2-yl)-5-methylcyclohexanol (**27a**)

Prepared with **25a** and imidazole at reflux for 12 h. Yield: 67%, white crystal, m.p. = 118–119 °C. ${\left[\alpha \right]}_{D}^{20}$ = +11.0 (c 0.30, MeOH). ^1^H NMR (500 MHz, CDCl_3_): δ = 0.89 (3H, d, *J* = 6.2 Hz), 0.92–1.02 (1H, m), 1.12 (1H, t, *J* = 12.2 Hz), 1.56–1.64 (2H, m), 1.74–1.90 (4H, m), 2.95 (1H, d, *J* = 9.4 Hz), 3.40 (1H, d, *J* = 9.4 Hz), 4.00 (1H, d, *J* = 13.9 Hz), 4.11 (1H, d, *J* = 13.9 Hz), 4.21 (1H, s), 4.37 (1H, d, *J* = 11.7 Hz), 4.48 (1H, d, *J* = 11.7 Hz), 6.90 (1H, s), 6.96 (1H, s), 7.25–7.37 (1H, s), 7.43 (1H, s). ^13^C NMR (125 MHz, CDCl_3_): δ = 20.0, 22.2, 25.8, 34.7, 42.5, 42.9, 51.4, 68.1, 70.3, 73.5, 75.5, 120.8, 128.1, 128.2, 128.6, 128.7, 137.7, 138.7. Found: C, 69.77; H, 8.15; N, 8.12. Anal. Calcd for C_20_H_28_N_2_O_3_: C, 69.74; H, 8.19; N, 8.13.

##### (1*S*,2*R*,5*R*)-2-((*R*)-1-(Benzyloxy)-2-hydroxy-3-(1H-imidazol-1-yl)propan-2-yl)-5-methylcyclohexanol (**27b**)

Prepared with **25b** and imidazole at reflux for 12 h. Yield: 83%, white crystal, m.p. = 149–150 °C. ${\left[\alpha \right]}_{D}^{20}$ = +20.0 (c 0.275, MeOH). ^1^H NMR (500 MHz, CDCl_3_): δ = 0.89 (3H, d, *J* = 6.2 Hz), 0.90–0.95 (1H, m), 1.15 (1H, td, *J* = 12.7, 2.0 Hz), 1.46–1.50 (1H, m), 1.62–1.65 (1H, m), 1.69–1.85 (4H, m), 2.81 (1H, d, *J* = 9.3 Hz), 3.24 (1H, d, *J* = 9.2 Hz), 4.17 (2H, q, *J* = 14.0 Hz), 4.35 (1H, d, *J* = 11.7 Hz), 4.37 (1H, d, *J* = 1.5 Hz), 4.46 (1H, d, *J* = 11.7 Hz), 6.94 (1H, s), 6.97 (1H, s), 7.29–7.37 (5H, m), 747 (1H, s). ^13^C NMR (125 MHz, CDCl_3_): δ = 20.3, 22.2, 25.8, 34.7, 42.6, 42.9, 52.4, 67.9, 69.3, 73.4, 75.8, 120.9, 128.0, 128.1, 128.5, 128.6, 137.7, 138.6. Found: C, 69.79; H, 8.22; N, 8.17. Anal. Calcd for C_20_H_28_N_2_O_3_: C, 69.74; H, 8.19; N, 8.13.

##### (1*S*,2*R*,5*R*)-2-((*S*)-1-(Benzyloxy)-2-hydroxy-3-(1H-1,2,4-triazol-1-yl)propan-2-yl)-5-methylcyclohexanol (**28a**)

Prepared with **25a** and 1,2,4-triazole at reflux for 12 h. Yield: 83%, colorless oil. ${\left[\alpha \right]}_{D}^{20}$ = +12.0 (c 0.30, MeOH). ^1^H NMR (500 MHz, CDCl_3_): δ = 0.88 (3H, d, *J* = 6.4 Hz), 0.92–0.99 (1H, m), 1.07 (1H, td, *J* = 12.0, 1.7 Hz), 1.61–1.66 (2H, m), 1.78–1.91 (4H, m), 0.86 (1H, d, *J* = 9.5 Hz), 2.87 (1H, s), 2.94 (1H, s), 3.46 (1H, d, *J* = 9.6 Hz), 4.22 (1H, s), 4.35–4.46 (4H, m), 7.25–7.37 (5H, m), 7.92 (1H, s), 8.02 (1H, s). ^13^C NMR (125 MHz, CDCl_3_): δ = 20.0, 22.2, 25.8, 34.9, 42.5, 43.4, 53.2, 67.5, 70.5, 73.7, 75.9, 128.1, 128.2, 1287, 137.5, 151.9. Found: C, 66.03; H, 7.90; N, 12.20. Anal. Calcd for C_19_H_27_N_3_O_3_: C, 66.06; H, 7.88; N, 12.16.

##### (1*S*,2*R*,5*R*)-2-((*R*)-1-(Benzyloxy)-2-hydroxy-3-(1H-1,2,4-triazol-1-yl)propan-2-yl)-5-methylcyclohexanol (**28b**)

Prepared with **25b** and 1,2,4-triazole at reflux for 12 h. Yield: 83%, colorless oil. ${\left[\alpha \right]}_{D}^{20}$ = +15.0 (c 0.25, MeOH). ^1^H NMR (500 MHz, CDCl_3_): δ = 0.88 (3H, d, *J* = 6.3 Hz), 0.87–0.95 (1H, m), 1.10 (1H, td, *J* = 12.6, 1.6 Hz), 1.22–1.29 (1H, m), 1.49–1.52 (1H, m), 1.60–1.64 (1H, m), 1.70–1.81 (2H, m), 1.81–1.87 (2H, m), 2.81 (1H, d, *J* = 9.6 Hz), 3.28 (1H, d, *J* = 9.6 Hz), 4.35–4.52 (5H, m), 7.25–7.36 (5H, m), 7.93 (1H, s), 8.05 (1H, s). ^13^C NMR (125 MHz, CDCl_3_): δ = 20.3, 22.2, 25.8, 34.9, 42.6, 43.6, 53.7, 67.6, 70.4, 73.7, 76.3, 128.1, 128.3, 128.7, 137.5. Found: C, 66.10; H, 7.85; N, 12.14. Anal. Calcd for C_19_H_27_N_3_O_3_: C, 66.06; H, 7.88; N, 12.16.

##### (*R*)-2-((1*R*,2*R*,4*R*)-2-(Benzyloxy)-4-methylcyclohexyl)-1-(1H-imidazol-1-yl)propan-2-ol (**31a**)

Prepared with **29** and imidazole at reflux for 96 h. Yield: 38%, colorless oil. ${\left[\alpha \right]}_{D}^{20}$ = −34.0 (c 0.20, MeOH). ^1^H NMR (500 MHz, CDCl_3_): δ = 0.80–1.01 (2H, m), 0.96 (3H, d, *J* = 6.1 Hz), 1.01 (3H, s), 1.10–1.20 (1H, m), 1.40–1.55 (1H, m), 1.72 (1H, d, *J* = 12.7 Hz), 1.97 (1H, d, *J* = 10.7 Hz), 2.30 (1H, d, *J* = 11.9 Hz), 3.46 (1H, t, *J* = 7.8 Hz), 4.18 (1H, s), 4.35 (1H, d, *J* = 10.9 Hz), 4.50 (1H, s), 4.76 (1H, d, *J* = 10.8 Hz), 7.02 (1H, s), 7.16 (1H, s), 7.31–7.39 (5H, m), 9.26 (1H, s). ^13^C NMR (125 MHz, CDCl_3_): δ = 22.1, 26.0, 31.6, 34.5, 40.1, 49.7, 57.6, 70.6, 73.4, 80.3, 118.2, 122.9, 128.6, 128.7, 129.0, 137.7. Found: C, 73.17; H, 8.60; N, 8.55. Anal. Calcd for C_20_H_28_N_2_O_2_: C, 73.14; H, 8.59; N, 8.53.

##### (*S*)-2-((1*R*,2*R*,4*R*)-2-(Benzyloxy)-4-methylcyclohexyl)-1-(1H-imidazol-1-yl)propan-2-ol (**31b**)

Prepared with **29** and imidazole at reflux for 96 h. Yield: 58%, white crystal, m.p = 170–172 °C.
${\left[\alpha \right]}_{D}^{20}$
= −48.0 (c 0.21, MeOH). ^1^H NMR (500 MHz, CDCl_3_): δ = 0.87–1.00 (2H, m), 0.95 (3H, d, *J* = 5.4 Hz), 1.05–1.16 (4H, m), 1.32 (1H, s), 1.43 (1H, s), 1.70–1.90 (2H, m), 2.29 (1H, d, *J* = 11.2 Hz), 3.62 (1H, s), 4.19 (1H, brs), 4.39 (1H, d, *J* = 10.9 Hz), 4.70 (1H, d, *J* = 10.9 Hz), 5.65 (1H, s), 7.28–7.38 (7H, m). ^13^C NMR (125 MHz, CDCl_3_): δ = 22.0, 27.5, 31.3, 34.2, 39.5, 48.2, 70.4, 74.4, 80.2, 128.5, 128.6, 128.9. Found: C, 73.10; H, 8.55; N, 8.57. Anal. Calcd for C_20_H_28_N_2_O_2_: C, 73.14; H, 8.59; N, 8.53.

##### (*R*)-2-((1*R*,2*R*,4*R*)-2-(Benzyloxy)-4-methylcyclohexyl)-1-(1H-1,2,4-triazol-1-yl)propan-2-ol (**32a**)

Prepared with **29** and 1,2,4-triazole at reflux for 24 h. Yield: 67%, colorless oil. ${\left[\alpha \right]}_{D}^{20}$ = −40.0 (c 0.265, MeOH). ^1^H NMR (500 MHz, CDCl_3_): δ = 0.85–0.94 (1H, m), 0.96 (3H, d, *J* = 6.6 Hz), 0.97 (3H, s), 1.05–1.13 (1H, m), 1.40–1.45 (1H, m), 1.50 (1H, td, *J* = 9.7, 3.1 Hz), 1.63 (1H, brs), 1.72 (1H, d, *J* = 13.0 Hz), 1.93 (1H, dd, *J* = 13.2, 3.2 Hz), 2.28 (1H, d, *J* = 12.1 Hz), 3.39 (1H, td, *J* = 10.5, 3.8 Hz), 4.12 (2H, q, *J* = 13.8 Hz), 4.26 (1H, d, *J* = 11.0 Hz), 4.72 (1H, d, *J* = 10.9 Hz), 4.98 (1H, s), 7.26–7.40 (5H, m), 7.88 (1H, s), 7.91 (1H, s). ^13^C NMR (125 MHz, CDCl_3_): δ = 22.1, 23.4, 26.5, 31.5, 34.6, 39.9, 50.8, 57.1, 70.3, 74.4, 80.3, 128.5, 128.6, 129.0, 137.5, 144.8, 151.1. Found: C, 69.32; H, 8.24; N, 12.80. Anal. Calcd for C_19_H_27_N_3_O_2_: C, 69.27; H, 8.26; N, 12.76.

##### (*S*)-2-((1*R*,2*R*,4*R*)-2-(Benzyloxy)-4-methylcyclohexyl)-1-(1H-1,2,4-triazol-1-yl)propan-2-ol (**32b**)

Prepared with **29** and 1,2,4-triazole at reflux for 24 h. Yield: 83%, colorless oil. ${\left[\alpha \right]}_{D}^{20}$ = −41.0 (c 0.285, MeOH). ^1^H NMR (500 MHz, CDCl_3_): δ = 0.78–0.87 (1H, m), 0.89–0.96 (1H, m), 0.93 (3H, d, *J* = 6.5 Hz), 0.97–1.04 (1H, m), 1.13 (3H, s), 1.15–1.21 (1H, m), 1.35–1.45 (1H, m), 1.62 (1H, s), 1.66 (1H, d, *J* = 13.1 Hz), 2.09 (1H, dd, *J* = 12.8, 3.1 Hz), 2.23 (1H, d, *J* = 12.1 Hz), 3.59 (1H, td, *J* = 10.4, 3.8 Hz), 3.99 (1H, d, *J* = 14.2 Hz), 4.26 (1H, d, *J* = 14.1 Hz), 4.39 (1H, d, *J* = 11.0 Hz), 4.67 (1H, d, *J* = 11.0 Hz), 5.46 (1H, s), 7.29–7.35 (5H, m), 7.87 (1H, s), 8.31 (1H, s). ^13^C NMR (125 MHz, CDCl_3_): δ = 21.5, 22.1, 27.0, 31.4, 34.0, 39.5, 47.4, 58.0, 70.2, 74.8, 80.4, 128.3, 128.4, 128.8, 137.2, 145.2, 150.7. Found: C, 69.25; H, 8.28; N, 12.73. Anal. Calcd for C_19_H_27_N_3_O_2_: C, 69.27; H, 8.26; N, 12.76.

##### (*S*)-2-((1*R*,2*R*,4*R*)-2-(Benzyloxy)-4-methylcyclohexyl)-3-(1H-imidazol-1-yl)propane-1,2-diol (**35a**)

Prepared with **33** and imidazole at reflux for 12 h. Yield: 67%, white crystal, m.p. = 135–136 °C. ${\left[\alpha \right]}_{D}^{20}$ = −42.0 (c 0.25, MeOH). ^1^H NMR (500 MHz, CDCl_3_): δ = 0.92–1.02 (2H, m), 0.96 (3H, d, *J* = 6.5 Hz), 1.03–1.09 (1H, m), 1.35–1.43 (1H, m), 1.69–1.72 (1H, m), 1.83–1.92 (3H, m), 2.27 (1H, d, *J* = 10.9 Hz), 3.21 (1H, d, *J* = 11.3 Hz), 3.32 (1H, td, *J* = 10.6, 4.0 Hz), 3.39 (1H, d, *J* = 11.2 Hz), 3.90 (2H, s), 4.21 (1H, d, *J* = 11.0 Hz), 4.64 (1H, d, *J* = 11.0 Hz), 6.93 (1H, s), 7.02 (1H, s), 7.30–7.39 (5H, m), 7.44 (1H, s). ^13^C NMR (125 MHz, CDCl_3_): δ = 22.1, 26.5, 31.3, 34.4, 40.1, 46.7, 50.5, 65.8, 70.0, 76.3, 79.7, 121.0, 128.5, 128.6, 128.9, 129.0, 137.2, 138.5. Found: C, 69.77; H, 8.17; N, 8.10. Anal. Calcd for C_20_H_28_N_2_O_3_: C, 69.74; H, 8.19; N, 8.13.

##### (*R*)-2-((1*R*,2*R*,4*R*)-2-(Benzyloxy)-4-methylcyclohexyl)-3-(1H-imidazol-1-yl)propane-1,2-diol (**35b**)

Prepared with **33** and imidazole at reflux for 12 h. Yield: 50%, colorless oil. ${\left[\alpha \right]}_{D}^{20}$ = −45.0 (c 0.185, MeOH). ^1^H NMR (500 MHz, CDCl_3_): δ = 0.78–0.87 (1H, m), 0.88–0.96 (1H, m), 0.92 (3H, d, *J* = 6.5 Hz), 1.13–1.19 (1H, m), 1.35–1.41 (2H, m), 1.67 (1H, d, *J* = 13.2 Hz), 1.75 (1H, dd, *J* = 13.1, 2.9 Hz), 1.85–2.10 (2H, m), 2.22 (1H, d, *J* = 12.3 Hz), 3.40 (1H, d, *J* = 11.2 Hz), 3.50 (1H, t, *J* = 11.1 Hz), 3.70 (1H, td, *J* = 10.5, 3.8 Hz), 3.90 (1H, d, *J* = 14.4 Hz), 4.06 (1H, d, *J* = 14.5 Hz), 4.37 (1H, d, *J* = 11.1 Hz), 4.66 (1H, d, *J* = 11.1 Hz), 7.01 (2H, s), 7.25–7.37 (5H, m), 7.54 (1H, s). ^13^C NMR (125 MHz, CDCl_3_): δ = 22.0, 26.7, 31.4, 34.2, 39.9, 45.6, 51.6, 64.7, 70.2, 76.5, 80.0, 121.0, 128.4, 128.5, 128.6, 128.9, 137.1, 138.7. Found: C, 69.73; H, 8.22; N, 8.17. Anal. Calcd for C_20_H_28_N_2_O_3_: C, 69.74; H, 8.19; N, 8.13.

##### (*S*)-2-((1*R*,2*R*,4*R*)-2-(Benzyloxy)-4-methylcyclohexyl)-3-(1H-1,2,4-triazol-1-yl)propane-1,2-diol (**36a**)

Prepared with **33** and 1,2,4-triazole at reflux for 12 h. Yield: 58%, colorless oil. ${\left[\alpha \right]}_{D}^{20}$ = −32.0 (c 0.26, MeOH). ^1^H NMR (500 MHz, CDCl_3_): δ = 0.88–1.01 (2H, m), 0.97 (3H, d, *J* = 6.5 Hz), 1.07–1.15 (1H, m), 1.39–1.46 (1H, m), 1.70–1.80 (2H, m), 1.95–2.00 (1H, m), 2.30 (1H, dd, *J* = 12.3, 1.5 Hz), 3.28 (2H, dd, *J* = 13.5, 12.2 Hz), 3.46 (1H, td, *J* = 10.5, 4.0 Hz), 4.03 (1H, d, *J* = 14.1 Hz), 4.27 (1H, d, *J* = 11.1 Hz), 4.28 (1H, d, *J* = 14.1 Hz), 4.70 (1H, d, *J* = 11.1 Hz), 7.33–7.42 (5H, m), 7.91 (1H, s), 7.92 (1H, s). ^13^C NMR (125 MHz, CDCl_3_): δ = 22.1, 26.1, 31.2, 34.5, 40.0, 48.2, 52.9, 66.1, 70.0, 76.1, 79.5, 128.7, 129.0, 137.2, 151.2. Found: C, 66.10; H, 7.89; N, 12.12. Anal. Calcd for C_19_H_27_N_3_O_3_: C, 66.06; H, 7.88; N, 12.16.

##### (*R*)-2-((1*R*,2*R*,4*R*)-2-(Benzyloxy)-4-methylcyclohexyl)-3-(1H-1,2,4-triazol-1-yl)propane-1,2-diol (**36b**)

Prepared with **33** and 1,2,4-triazole at reflux for 12 h. Yield: 50%, colorless oil. ${\left[\alpha \right]}_{D}^{20}$ = −32.0 (c 0.24, MeOH). ^1^H NMR (500 MHz, CDCl_3_): δ = 0.79–0.97 (3H, m), 0.93 (3H, d, *J* = 6.5 Hz), 1.07–1.17 (1H, m), 1.26 (2H, s), 1.28–1.43 (4H, m), 1.69 (1H, d, *J* = 13.2 Hz), 2.06–2.09 (1H, m), 2.23 (1H, d, *J* = 12.2 Hz), 3.43–3.49 (2H, m), 3.68 (1H, td, *J* = 10.5, 3.9 Hz), 4.23 (1H, d, *J* = 14.4 Hz), 4.35 (1H, d, *J* = 14.3 Hz), 4.36 (1H, d, *J* = 11.0 Hz), 4.66 (1H, d, *J* = 11.1 Hz), 5.50 (1H, brs), 7.25–7.37 (5H, m), 7.91 (1H, s), 8.23 (1H, s). ^13^C NMR (125 MHz, CDCl_3_): δ = 22.0, 26.4, 29.8, 31.3, 34.3, 39.9, 46.2, 54.3, 64.4, 70.2, 76.4, 80.0, 128.4, 128.5, 128.9, 137.1, 150.6. Found: C, 66.03; H, 7.92; N, 12.18. Anal. Calcd for C_19_H_27_N_3_O_3_: C, 66.06; H, 7.88; N, 12.16.

##### (*S*)-1-(Benzyloxy)-2-((1*R*,2*R*,4*R*)-2-(benzyloxy)-4-methylcyclohexyl)-3-(1H-imidazol-1-yl)propan-2-ol (**39a**)

Prepared with **35a** and imidazole at reflux for 48 h. Yield: 67%, colorless oil. ${\left[\alpha \right]}_{D}^{20}$ = −72.0 (c 0.28, MeOH). ^1^H NMR (500 MHz, CDCl_3_): δ = 0.86–1.10 (3H, m), 0.96 (3H, d, *J* = 6.5 Hz), 1.37–1.42 (1H, m), 1.69 (1H, d, *J* = 12.6 Hz), 1.79 (1H, td, *J* = 12.6, 3.2 Hz), 2.00–2.04 (2H, m), 2.28 (1H, d, *J* = 12.1 Hz), 3.13 (2H, d, *J* = 8.9 Hz), 3.39 (1H, td, *J* = 10.6, 3.9 Hz), 3.83 (1H, d, *J* = 13.9 Hz), 4.07 (1H, d, *J* = 14.0 Hz), 4.24 (1H, d, *J* = 11.0 Hz), 4.33 (1H, d, *J* = 11.8 Hz), 4.43 (1H, d, *J* = 11.8 Hz), 4.70 (1H, d, *J* = 11.1 Hz), 4.82 (1H, brs), 6.90 (1H, s), 7.00 (1H, s), 7.25–7.37 (10H, m), 7.47 (1H, s). ^13^C NMR (125 MHz, CDCl_3_): δ = 22.1, 26.4, 31.4, 34.6, 40.0, 49.0, 51.4, 70.2, 73.6, 75.6, 79.8, 121.4, 127.9, 128.6, 128.9, 137.3, 137.9. Found: C, 74.65; H, 7.93; N, 6.48. Anal. Calcd for C_27_H_34_N_2_O_3_: C, 74.62; H, 7.89; N, 6.45.

##### (*R*)-1-(Benzyloxy)-2-((1*R*,2*R*,4*R*)-2-(benzyloxy)-4-methylcyclohexyl)-3-(1H-imidazol-1-yl)propan-2-ol (**39b**)

Prepared with **35a** and imidazole at reflux for 48 h. Yield: 83%, colorless oil. ${\left[\alpha \right]}_{D}^{20}$ = −48.0 (c 0.285, MeOH). ^1^H NMR (500 MHz, CDCl_3_): δ = 0.76–0.95 (3H, m), 0.93 (1H, d, *J* = 6.5 Hz), 1.38–1.50 (3H, m), 1.65 (1H, d, *J* = 13.2 Hz), 1.73 (1H, d, *J* = 10.2 Hz), 1.96 (2H, brs), 2.20 (1H, d, *J* = 12.3 Hz), 3.11 (1H, d, *J* = 9.7 Hz), 3.32 (1H, d, *J* = 9.7 Hz), 3.54 (1H, td, *J* = 10.2, 3.8 Hz), 3.96 (1H, d, *J* = 14.0 Hz), 4.12 (1H, d, *J* = 14.1 Hz), 4.16 (1H, d, *J* = 11.0 Hz), 4.34 (1H, d, *J* = 11.9 Hz), 4.49 (1H, d, *J* = 11.9 Hz), 4.56 (1H, d, *J* = 11.0 Hz), 5.17 (1H, s), 7.00 (1H, s), 7.01 (1H, s), 7.21–7.38 (10H, m), 7.55 (1H, s). ^13^C NMR (125 MHz, CDCl_3_): δ = 22.1, 26.7, 31.4, 34.6, 39.8, 48.1, 52.2, 70.1, 72.1, 73.7, 75.3, 80.5, 121.3, 128.2, 128.6, 128.8, 137.2, 137.9, 138.7. Found: C, 74.60; H, 7.87; N, 6.50. Anal. Calcd for C_27_H_34_N_2_O_3_: C, 74.62; H, 7.89; N, 6.45.

##### (*S*)-1-(Benzyloxy)-2-((1*R*,2*R*,4*R*)-2-(benzyloxy)-4-methylcyclohexyl)-3-(1H-1,2,4-triazol-1-yl)propan-2-ol (**40a**)

Prepared with **35a** and 1,2,4-triazole at reflux for 48 h. Yield: 83%, colorless oil. ${\left[\alpha \right]}_{D}^{20}$ = −58.0 (c 0.265, MeOH). ^1^H NMR (500 MHz, CDCl_3_): δ = 0.84–0.97 (2H, m), 0.94 (3H, d, *J* = 6.5 Hz), 1.04–1.13 (1H, m), 1.38–1.45 (1H, m), 1.67 (1H, d, *J* = 12.8 Hz), 1.78 (1H, td, *J* = 12.5, 3.2 Hz), 1.90–1.94 (1H, m), 2.25 (1H, d, *J* =12.3 Hz), 2.59 (2H, s), 3.18 (1H, d, *J* = 10.0 Hz), 3.32 (1H, d, *J* = 10.0 Hz), 3.43 (1H, td, *J* = 10.6, 3.9 Hz), 4.18 (1H, d, *J* = 14.1 Hz), 4.20 (1H, d, *J* = 10.9 Hz), 4.33 (1H, d, *J* = 14.0 Hz), 4.40 (1H, d, *J* = 11.9 Hz), 4.50 (1H, d, *J* = 11.9 Hz), 4.64 (1H, d, *J* = 10.9 Hz), 5.02 (1H, brs), 7.25–7.36 (10H, m), 7.88 (1H, s), 7.97 (1H, s). ^13^C NMR (125 MHz, CDCl_3_): δ = 22.1, 26.1, 31.4, 34.5, 40.1, 47.7, 54.3, 70.1, 73.1, 73.8, 75.9, 79.5, 127.9, 128.0, 128.4, 128.5, 128.6, 128.9, 137.5, 138.0, 150.5. Found: C, 71.69; H, 7.67; N, 9.66. Anal. Calcd for C_26_H_33_N_3_O_3_: C, 71.70; H, 7.64; N, 9.65.

##### (*R*)-1-(Benzyloxy)-2-((1*R*,2*R*,4*R*)-2-(benzyloxy)-4-methylcyclohexyl)-3-(1H-1,2,4-triazol-1-yl)propan-2-ol (**40b**)

Prepared with **35a** and 1,2,4-triazole at reflux for 48 h. Yield: 83%, colorless oil. ${\left[\alpha \right]}_{D}^{20}$ = −57.0 (c 0.265, MeOH). ^1^H NMR (500 MHz, CDCl_3_): δ = 0.74–0.95 (3H, m), 0.91 (3H, d, *J* = 6.5 Hz), 1.17–1.42 (6H, m), 1.64 (1H, d, *J* = 13.7 Hz), 2.06 (1H, d, *J* = 10.1 Hz), 2.18 (1H, d, *J* = 12.2 Hz), 3.28 (1H, d, *J* = 9.8 Hz), 3.41 (1H, d, *J* = 9.7 Hz), 3.59 (1H, td, *J* = 10.2, 3.8 Hz), 4.19 (1H, d, *J* = 10.9 Hz), 4.27 (1H, d, *J* = 14.3 Hz), 4.40 (1H, d, *J* = 14.3 Hz), 4.41 (1H, d, *J* = 12.0 Hz), 4.55 (1H, d, *J* = 11.9 Hz), 456 (1H, d, *J* = 10.9 Hz), 5.35 (1H, s), 7.21–7.37 (10H, m), 7.89 (1H, s), 8.28 (1H, s). ^13^C NMR (125 MHz, CDCl_3_): δ = 22.1, 26.6, 31.4, 34.5, 39.9, 47.4, 54.8, 70.1, 72.6, 73.8, 75.8, 80.4, 128.0, 128.1, 128.2, 128.4, 128.7, 137.3, 138.0, 150.7. Found: C, 71.73; H, 7.60; N, 9.62. Anal. Calcd for C_26_H_33_N_3_O_3_: C, 71.70; H, 7.64; N, 9.65.

##### (1*R*,2*R*,5*R*)-2-((*S*)-1-(Benzyloxy)-2-hydroxy-3-(1H-imidazol-1-yl)propan-2-yl)-5-methylcyclohexanol (**42a**)

Prepared with **35b** and imidazole at reflux for 12 h. Yield: 58%, white crystal, m.p. = 133–134 °C. ${\left[\alpha \right]}_{D}^{20}$ = −22.0 (c 0.25, MeOH). ^1^H NMR (500 MHz, CDCl_3_): δ = 0.84–0.92 (1H, m), 0.92 (3H, d, *J* = 6.5 Hz), 0.97–1.07 (2H, m), 1.39–1.47 (1H, m), 1.65–1.77 (2H, m), 1.84–1.88 (1H, m), 1.94–1.97 (H, m), 3.16 (1H, d, *J* = 9.5 Hz), 3.28 (1H, d, *J* = 9.5 Hz), 3.64 (1H, td, *J =* 10.5, 4.2 Hz), 4.11 (1H, d, *J* = 14.3 Hz), 4.20 (1H, d, *J* = 14.2 Hz), 4.38 (1H, d, *J* = 11.8 Hz), 4.48 (1H, d, *J* = 11.8 Hz), 7.00 (2H, s), 7.25–7.35 (5H, m), 7.53 (1H, s). ^13^C NMR (125 MHz, CDCl_3_): δ = 21.9, 26.2, 31.4, 34.6, 45.5, 49.7, 50.6, 72.0, 73.3, 73.7, 76.5, 121.2, 127.9, 128.1, 128.6, 128.7, 137.7, 138.8. Found: C, 69.71; H, 8.16; N, 8.15. Anal. Calcd for C_20_H_28_N_2_O_3_: C, 69.74; H, 8.19; N, 8.13.

##### (1*R*,2*R*,5*R*)-2-((*R*)-1-(Benzyloxy)-2-hydroxy-3-(1H-imidazol-1-yl)propan-2-yl)-5-methylcyclohexanol (**42b**)

Prepared with **35b** and imidazole at reflux for 12 h. Yield: 58%, colorless oil. ${\left[\alpha \right]}_{D}^{20}$ = −8.0 (c 0.25, MeOH). ^1^H NMR (500 MHz, CDCl_3_): δ = 0.77 (1H, q, *J* = 11.6 Hz), 0.89 (3H, d, *J* = 6.4 Hz), 0.99 (1H, d, *J* = 11.6 Hz), 1.26–1.42 (3H, m), 1.64 (1H, d, *J* = 13.3 Hz), 1.69 (1H, d, *J* = 8.5 Hz), 1.91 (1H, d, *J* = 12.1 Hz), 3.38 (1H, d, *J* = 9.7 Hz), 3.57 (1H, d, *J* = 9.6 Hz), 3.80 (1H, t, *J* = 7.4 Hz), 4.02 (1H, d, *J* = 14.2 Hz), 4.16 (1H, d, *J* = 14.2 Hz), 4.49 (1H, d, *J* = 11.9 Hz), 4.55 (1H, d, *J* = 11.8 Hz), 6.98 (1H, s), 6.99 (1H, s), 7.25–7.38 (5H, m), 7.52 (1H, s). ^13^C NMR (125 MHz, CDCl_3_): δ = 21.9, 26.4, 31.5, 34.5, 45.3, 48.3, 52.4, 72.2, 72.5, 73.9, 76.3, 121.1, 127.9, 128.1, 128.3, 128.7, 137.8, 138.7. Found: C, 69.77; H, 8.17; N, 8.10. Anal. Calcd for C_20_H_28_N_2_O_3_: C, 69.74; H, 8.19; N, 8.13.

##### (1*R*,2*R*,5*R*)-2-((*S*)-1-(Benzyloxy)-2-hydroxy-3-(1H-1,2,4-triazol-1-yl)propan-2-yl)-5-methylcyclohexanol (**43a**)

Prepared with **35b** and 1,2,4-triazole at reflux for 12 h. Yield: 67%, white crystal, m.p. = 53–54 °C. ${\left[\alpha \right]}_{D}^{20}$ = −16.0 (c 0.25, MeOH). ^1^H NMR (500 MHz, CDCl_3_): δ = 0.80–0.89 (1H, m), 0.91 (1H, d, *J* = 6.5 Hz), 0.96–1.07 (2H, m), 1.21–1.46 (3H, m), 1.62–1.68 (2H, m), 181–1.85 (1H, m), 1.95–1.99 (1H, m), 3.27 (1H, d, *J* = 9.7 Hz), 3.34 (1H, d, *J* = 9.7 Hz), 3.58 (1H, td, *J* = 10.6, 4.1 Hz), 4.37–4.53 (4H, m), 7.25–7.36 (5H, m), 7.91 (1H, s), 8.17 (1H, s). ^13^C NMR (125 MHz, CDCl_3_): δ = 22.0, 25.8, 31.4, 34.5, 45.4, 48.9, 53.6, 71.7, 72.8, 73.8, 76.5, 127.9, 128.1, 128.6, 137.5, 151.3. Found: C, 66.10; H, 7.85; N, 12.12. Anal. Calcd for C_19_H_27_N_3_O_3_: C, 66.06; H, 7.88; N, 12.16.

##### (1*R*,2*R*,5*R*)-2-((*R*)-1-(Benzyloxy)-2-hydroxy-3-(1H-1,2,4-triazol-1-yl)propan-2-yl)-5-methylcyclohexanol (**43b**)

Prepared with **35b** and 1,2,4-triazole at reflux for 12 h. Yield: 58%, colorless oil. ${\left[\alpha \right]}_{D}^{20}$ = −6.0 (c 0.25, MeOH). ^1^H NMR (500 MHz, CDCl_3_): δ = 0.71–0.79 (1H, m), 0.88 (1H, d, *J* = 6.5 Hz), 0.93 (1H, q, *J* = 12.0 Hz), 1.26–1.29 (2H, m), 1.36–1.43 (1H, m), 1.63 (1H, d, *J* = 13.2 Hz), 1.88–1.94 (2H, m), 3.45 (1H, d, *J* = 9.8 Hz), 3.61 (1H, d, *J* = 9.8 Hz), 3.79 (1H, td, *J* = 10.4, 4.1 Hz), 4.37 (1H, d, *J* = 14.3 Hz), 4.41 (1H, d, *J* = 14.3 Hz), 4.50 (1H, d, *J* = 11.9 Hz), 4.56 (1H, d, *J* = 11.9 Hz), 7.26–7.37 (5H, m), 7.89 (1H, s), 8.21 (1H, s). ^13^C NMR (125 MHz, CDCl_3_): δ = 21.9, 26.2, 31.4, 34.5, 45.2, 48.2, 54.7, 72.2, 74.0, 127.9, 128.1, 128.6, 137.7, 145.0, 151.1. Found: C, 66.03; H, 7.90; N, 12.19. Anal. Calcd for C_19_H_27_N_3_O_3_: C, 66.06; H, 7.88; N, 12.16.

#### 4.2.7. General Procedure for Debenzylation

A suspension of palladium-on-carbon (5% Pd/C, 0.22 g) in MeOH (50 mL) was added to (+)-neoisopulegol-based *O*-benzyl derivatives (14.0 mmol) in MeOH (100 mL) and the mixture was stirred under a H_2_ atmosphere (1 atm) at room temperature. After completion of the reaction (as monitored by TLC, 24 h), the mixture was filtered through a Celite pad and the solution was evaporated to dryness. The crude products were recrystallized in diethyl ether, resulting in primary aminodiols (**9a**–**b**) and aminotriols (**16a**–**b**).

##### (1*S*,2*R*,5*R*)-2-((*R*)-1-Amino-2-hydroxypropan-2-yl)-5-methylcyclohexanol (**9a**)

Prepared with **5a**. Yield: 91%, white crystal, m.p. = 100–110 °C ${\left[\alpha \right]}_{D}^{20}$ = +14.0 (c 0.25, MeOH). ^1^H NMR (500 MHz, DMSO-*d_6_*): δ = 0.75–0.85 (2H, m), 0.80 (3H, d, *J* = 5.3 Hz), 0.99 (1H, d, *J* = 12.1 Hz), 1.17 (3H, s), 1.31 (1H, d, *J* = 11.7 Hz), 1.45 (1H, q, *J* = 10.9 Hz), 1.58 (1H, d, *J* = 10.4 Hz), 1.65–1.80 (3H, m), 2.70 (1H, d, *J* = 12.7 Hz), 2.89 (1H, d, *J* = 12.7 Hz), 4.04 (1H, s), 4.95 (1H, brs). ^13^C NMR (125 MHz, DMSO-*d_6_*): δ = 20.1, 22.2, 23.3, 25.4, 34.7, 42.8, 45.9, 49.0, 65.1, 71.3. Found: C, 64.09; H, 11.35; N, 7.50. Anal. Calcd for C_10_H_21_NO_2_: C, 64.13; H, 11.30; N, 7.48.

##### (1*S*,2*R*,5*R*)-2-((*S*)-1-Amino-2-hydroxypropan-2-yl)-5-methylcyclohexanol (**9b**)

Prepared with **5b**. Yield: 91%, white crystal, m.p. = 138–140 °C. ${\left[\alpha \right]}_{D}^{20}$ = +10.0 (c 0.25, MeOH). ^1^H NMR (500 MHz, DMSO–*d_6_*): δ = 0.82 (3H, d, *J* = 5.7 Hz), 0.81–0.88 (1H, m), 1.02 (1H, t, *J* = 12.5 Hz), 1.17 (3H, s), 1.32 (1H, d, *J* = 10.2 Hz), 1.45–1.55 (2H, m), 1.65–1.80 (3H, m), 2.62 (1H, d, *J* = 12.7 Hz), 2.91 (1H, d, *J* = 12.8 Hz), 4.12 (1H, s), 4.86 (1H, brs), 6.85 (3H, brs). ^13^C NMR (125 MHz, DMSO-*d_6_*): δ = 20.7, 22.0, 25.1, 25.2, 34.7, 42.4, 45.2, 49.5, 64.3, 70.9. Found: C, 64.15; H, 11.27; N, 7.45. Anal. Calcd for C_10_H_21_NO_2_: C, 64.13; H, 11.30; N, 7.48.

##### (*S*)-3-Amino-2-((1*R*,2*S*,4*R*)-2-hydroxy-4-methylcyclohexyl)propane-1,2-diol (**16a**)

Prepared with **12a**, **20a** or **25a**. Yield: 78% (**12a**), 94% (**20a**), 91% (**25a**), white crystal, m.p. = 107–106 °C. ${\left[\alpha \right]}_{D}^{20}$ = +18.0 (c 0.30, MeOH). ^1^H NMR (500 MHz, DMSO-*d_6_*): δ = 0.80 (3H, d, *J* = 6.6 Hz), 0.79–0.90 (1H, m), 0.93–1.00 (1H, m), 1.38–1.41 (1H, m), 1.45–1.54 (2H, m), 1.60–1.70 (2H, m), 1.73–1.85 (1H, m), 2.60 (1H, d, *J* = 12.6 Hz), 3.30 (2H, q, *J* = 10.9 Hz), 4.07 (1H, s). ^13^C NMR (125 MHz, DMSO-*d_6_*): δ = 20.3, 22.4, 25.4, 35.1, 42.3, 44.1, 45.2, 64.4, 65.2, 74.9. Found: C, 59.10; H, 10.38; N, 6.93. Anal. Calcd for C_10_H_21_NO_3_: C, 59.08; H, 10.41; N, 6.89.

##### (*R*)-3-Amino-2-((1*R*,2*S*,4*R*)-2-hydroxy-4-methylcyclohexyl)propane-1,2-diol (**16b**)

Prepared with **20b** or **25b**. Yield: 94% (**20b**), 91% (**25b**), white crystal, m.p. = 80–82 °C. ${\left[\alpha \right]}_{D}^{20}$ = +13.0 (c 0.30, MeOH). ^1^H NMR (500 MHz, DMSO-*d_6_*): δ = 0.80 (3H, d, *J* = 6.6 Hz), 0.82–0.88 (1H, m), 0.94–0.99 (1H, m), 1.44–1.57 (3H, m), 1.64–1.69 (2H, m), 1.73–1.77 (1H, m), 2.57 (2H, q, *J* = 12.7 Hz), 3.32 (1H, d, *J* = 11.0 Hz), 3.39 (1H, d, *J* = 11.0 Hz), 4.00 (1H, s). ^13^C NMR (125 MHz, DMSO-*d_6_*): δ = 19.8, 22.4, 25.4, 35.1, 42.4, 44.7, 45.3, 64.2, 64.8, 75.3. Found: C, 59.05; H, 10.43; N, 6.87. Anal. Calcd for C_10_H_21_NO_3_: C, 59.08; H, 10.41; N, 6.89.

### 4.3. General Procedure for Antimicrobial Assays

For the antimicrobial analyses the pure compounds were first dissolved in MeOH and diluted with H_2_O to two concentration levels (400 µg mL^−1^ and 40 µg mL^−1^) keeping the final MeOH content at 10%. Then these solutions were investigated in microdilution assay with two Gram-positive bacteria including *Bacillus subtilis* SZMC 0209 and *Staphylococcus aureus* SZMC 14611, two Gram-negative bacteria *Escherichia coli* SZMC 6271 and *Pseudomonas aeruginosa* SZMC 23290, as well as two yeast strains *Candida albicans* SZMC 1533 and *C. krusei* SZMC 1352 according to the M07-A10 CLSI guideline [92] and our previous work [93]. Suspensions of the test microbes were prepared from overnight cultures cultivated in ferment broth (bacteria: 10 g L^−1^ peptone, 5 g L^−1^ NaCl, 5 g L^−1^ yeast extract; yeast: 20 g L^−1^ peptone, 10 g L^−1^ yeast extract, 20 g L^−1^ glucose) at 37 °C. Then the concentrations of the suspensions were set to 2 × 10^5^ cells mL^−1^ with sterile media. For the assay, 96-well plates were prepared by dispensing into each well 100 μL of suspension containing the bacterial or yeast cells and 50 μL of sterile broth as well as 50 μL of the test solutions and incubated for 24 h at 37 °C. The mixture of 150 μL broth and 50 μL of 10% methanol was used as the blank sample for the background correction, while 100 μL of microbial suspension supplemented with 50 μL sterile broth and 50 μL of 10% methanol was applied as negative control. The positive control contained ampicillin (Sigma) or nystatin (Sigma) for bacteria or fungi, respectively, at two final concentration levels (100 µg mL^−1^ and 10 µg mL^−1^). The inhibitory effects of the derivatives were observed spectrophotometrically at 620 nm after the incubation, and inhibition was calculated as the percentage of the positive control after blank correction.

The MIC was also determined for certain compounds, which were based on the broth microdilution method described above and in the M07-A10 CLSI guideline [92]. The compounds were prepared in two-fold dilutions in 10% MeOH covering the final concentration range of 0.78–100.00 µg/mL. The MIC was observed as the lowest concentration level of the compound that completely inhibits the growth of the organism in microdilution wells as detected by the unaided eye. All experiments were repeated three times.

## 5. Conclusions

The results of the present study establishing antimicrobial and antifungal behavior of some synthetic derivatives are promising with respect to possible clinical application. It is strongly believed that it will serve a suitable basis for future research on developing alternative antibiotics focusing on the development of better antibiotics against infectious organisms. The obtained results indicate that the di-*O*-benzyl derivatives may have considerable potential for therapeutic application as novel drug candidates against bacterial and fungal infections. Based on the results obtained, some of the studied compounds have proved to be promising candidates for additional efficacy evaluation.

Furthermore, in vitro studies have clearly shown that the *O*-benzyl substituent on the cyclohexyl ring in aminodiol and aminotriol derivatives is essential to have an antimicrobial effect whereas the stereochemistry of the *O*-benzyl substituent on the cyclohexane ring in the aminodiol and aminotriol function has no influence on the antimicrobial effect.

In addition, the antifungal activity was found to be affected by the stereochemistry of the derivatives, namely the *S*-isomers were more potent than the corresponding *R*-isomers against fungi while the antibacterial effect did not distinguish between the different stereoisomers.

In the next stage of our project, we plan to obtain *N*-benzyl and imidazole *O*-benzyl analogs, preferably different substitutions on *N*-benzyl and imidazole systems, to increase their antimicrobial activities on various microorganisms. For the optimized compounds, additionally, docking studies and molecular dynamics study will also be performed to get an insight into the dynamics of ligand interaction.

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
