# Peer review of "Novel (+)-Neoisopulegol-Based O-Benzyl Derivatives as Antimicrobial Agents"

_ijms, 2021, doi:10.3390/ijms22115626_

Round 1

Reviewer 1 Report

Szakonyi et al reported the synthesis of a library of (-) and (+)neoisopulegol-based O-benzyl derivatives of aminodiols and aminotriols starting from a commerially and enantiopure (–)-isopulegol. The synthesized compounds were screened for their antibacterial (Gram +ve and -ve) and antifungal (two strains) activities. The results led to identification of the critical structural and stereochemical features in the isopulegol-aminol scaffold which are critical for potential antimicrobial activity. Accordingly, a new series of neoisopulegol-based O-benzyl compounds was identified which exhibits a potential antimicrobial activity compared to reference drugs. This is a very interesting study which represents an advanced chemical aspect about the imidazole-containing compounds and isopulegol. The study is well designed and well performed. Nevertheless, there are some points that the authors should address before the manuscript would be suitable for publication. The major concerns in the presenting study are the determination of the IC50 of the most active compds and the prove of stereo-purity of compds. Please, carefully check the following comments:

1- the introduction part is well written, but please affiliate your intro with structures of the potent antimicrobial compds. It would be also of advantage to add few of your previous study (71,72) which is the motive force to do this work.

2- scheme 1 please add R referred to what?

3- the authors should briefly discuss in the synthetic part the description of the reaction performed regarding the stereochemistry and the role of reagents used. e.g, the role of LiClO4, Na2HPO4. 12H2O, KI, why you use LiClO4 (1eu) for benyl amine and used K2CO3 (5eq) for imidazole and triazol.

4- line 98: During our attempt to improve the resolution of epoxides, What does it mean?

5- line 99:The synthesis of 18a starting from 10, however, failed. Which trials you have done?

6- line 103; The reactions proceeded smoothly giving the corresponding 18a,b in good yields. Please remove it. its just repeating the same info.

7- for conversion of compd 3 to 10, what is the role of SeO2? please add a reference? epoxidation of the vinyl would be possible?

8- in scheme 5, the starting compd is 1 not 2

9- in scheme 6, conversion of compd 25a and 19a to 45 was performed by only hydrogenation? does hydrogenation also open the epoxide ring?

10- in table 1, what is RSD inhibitory effect?

11- In line 216; the enantiomeric purity of starting material 1 were performed GC measurements on a Chirasil-DEX CB column (2500 × 0.25 mm I.D.), please provide the spectra in the supporting materials.

12- Regarding the biological activity evaluation, the authors have screened the antimicrobial activity of synthesized compds at only to conc. 10ug/mL and 100ug/mL, which is logic for having an overview about the activity or SAR. Nevertheless, the authors should focus/select on the best compds and evaluate the IC50 for the most active/promising compds and give a final conclusion with the activity of the most promising compds. This would give a final message of the current study and would achieve the aim of the study. Otherwise, it's just huge piece of synthetic work without any significant biological outcome. In this regards, I suggest the following evaluation:

compds with potential antifungal activity: 10, 12a, 12b, 20b, 22b,27b, 31b, 39b, 43b in comparison with nystatin (C. albicans).

compds with potential antibacterial G+ve: 3, 5a, 5b, 7b, 10, 12a, 12b, 31a, 20, 22-23, 31b, 39b (B. subtilis)

compds with potential antibacterial G-ve: 5b, 36b (P.aerug)

line 165, it should be bactericidal activities.

13- I suggest to move the part from line 163-192 to the discussion part.

14- I'm missing the conclusion part and future perspectives about the current study.

15- In the discussion part about the evaluation of compds, it should be more deep discussion. It is too general, better to take insight each group of synthesized compds. For example, the stereoconfiguration effect about the two chiral centers, the (+) and (-) isopulegol, the 1ry O-Benzyl protected compds or 2ry O-benzyl, the O-benzyl or N-benzyl, the aminodiol and aminotriol.

16- Regarding the characterization of compds, the authors should provide the optical activity /HPLC (chiral) of synthesized compds, since  the stereo-purity of compounds would NOTg be proved by NMR analysis.

17- please provide better HNMR resolution for compd 12b,31b.

Author Response

Dear reviewer,

Please see the attachment with the detailed answer!

Sincerely Yours

Zsolt Szakonyi (corresponding author)

Reviewer 2 Report

The manuscript ijms-1215661 "Novel (+)-neoisopulegol-based O-benzyl derivatives as antimicrobial agents" by Szakonyi and co-workers describes the synthesis of new series of  (+)-neoisopulegol-based O-benzyl derivatives of aminodiols and aminotriols and the study the study their antimicrobial activity against bacterial and fungal strains. The synthetic part of the manuscript is well done and well written.

Comments and remarks:

1) The "Discussion" part of the manuscript is rather small. Authors should add information on structure-activity relationships and possible reasons for obtained results.

2) What is the mechanism of biological activity of the obtained compound against bacteria and fungi? How can the authors confirm it?

3) How can the authors explain the results "the R-isomers was more potent than the corresponding S-isomers against fungi"?

4) The study does not assess the impact of test compounds on normal cells or RBC assay.

5) I recommend the authors to make a separate section "Conclusions".

6) There are typos and inconvenient expressions in the text of the manuscript. Please check again.

Author Response

(The authors gave the same response as above.)

Round 2

Reviewer 1 Report

I really appreciate the efforts that the authors put to improve the manuscript. Thanks!

I think the manuscript has been significantly improved and accordingly I would recommend the publication of this interesting study in the present form.

Reviewer 2 Report

The authors have improved the manuscript. Recommended changes have been added.